# A Reproducible and Realistic Evaluation of Partial Domain Adaptation Methods

**Tiago Salvador**[†]  
*Mila - Quebec AI Institute, McGill University*

*tiago.saldanhasalvador@mail.mcgill.ca*

**Kilian Fatras**[†]  
*Mila - Quebec AI Institute, McGill University*

*kilian.fatras@mila.quebec*

**Ioannis Mitliagkas**  
*Mila - Quebec AI Institute, Université de Montréal, Canada CIFAR AI Chair*

*ioannis@mila.quebec*

**Adam Oberman**  
*Mila - Quebec AI Institute, McGill University, Canada CIFAR AI Chair*

*adam.oberman@mila.quebec*

**Reviewed on OpenReview:** *https://openreview.net/forum?id=XcVzIBXeRn*

## Abstract

Unsupervised Domain Adaptation (UDA) aims at classifying unlabeled target images leveraging source labeled ones. In the case of an extreme label shift scenario between the source and target domains, where we have extra source classes not present in the target domain, the UDA problem becomes a harder problem called Partial Domain Adaptation (PDA). While different methods have been developed to solve the PDA problem, most successful algorithms use model selection strategies that rely on target labels to find the best hyper-parameters and/or models along training. These strategies violate the main assumption in PDA: only unlabeled target domain samples are available. In addition, there are also experimental inconsistencies between developed methods - different architectures, hyper-parameter tuning, number of runs - yielding unfair comparisons. The main goal of this work is to provide a realistic evaluation of PDA methods under different model selection strategies and a consistent evaluation protocol. We evaluate 6 state-of-the-art PDA algorithms on 2 different real-world datasets using 7 different model selection strategies. Our two main findings are: *(i)* without target labels for model selection, the accuracy of the methods decreases up to 30 percentage points; *(ii)* only one method and model selection pair performs well on both datasets. Experiments were performed with our PyTorch framework, BenchmarkPDA, which we open source.

## 1 Introduction

**Domain adaptation.** Deep neural networks are highly successful in image recognition for in-distribution samples (He et al., 2016) with this success being intrinsically tied to the large number of labeled training data. However, they tend to not generalize as well on images with different backgrounds or colors not seen during training. These differences constitute some examples of what is known in the literature as covariate shift. Unfortunately, enriching the training set with new samples from different domains is challenging as labeling data is both an expensive and time-consuming task. Thus, researchers have focused on unsupervised domain adaptation (UDA) where we have access to unlabelled samples from a different domain, known as the target domain. The purpose of UDA is to classify these unlabeled samples by leveraging the knowledge given by the labeled samples from the source domain (Pan & Yang, 2010; Patel et al., 2015). In the standard

---

[†]These authors contributed equally to this work

UDA problem, the source and target domains are assumed to share the same classes. In this paper, we consider a more challenging variant of the problem called *partial domain adaptation* (PDA): the classes in the target domain $\mathcal{Y}_t$ form a subset of the classes in the source domain $\mathcal{Y}_s$ (Cao et al., 2018), *i.e.,* $\mathcal{Y}_t \subset \mathcal{Y}_s$. The number of target classes is unknown as we do not have access to the labels. The extra source classes, not present in the target domain, make the PDA problem more difficult: simply aligning the source and target domains results in matching target samples to source samples whose classes are not present in the target domain.

**Realistic evaluations.** Most recent PDA methods report an increase of the target accuracy up to 15 percentage points on average when compared to the baseline approach that is trained only on source samples. While these successes constitute important breakthroughs in the DA research literature, target labels are used for model selection, violating the main UDA assumption. In their absence, the effectiveness of PDA methods remains unclear and model selection still constitutes an open problem as we show in this work. Moreover, the hyper-parameter tuning is either unknown or lacks details and sometimes requires labeled target data, which makes it challenging to apply PDA methods to new datasets. Recent work has highlighted the importance of model selection in the presence of domain shift. Gulrajani & Lopez-Paz (2021) showed that when evaluating domain generalization (DG) algorithms, whose goal is to generalize to a completely unseen domain, in a consistent and realistic setting no method outperforms the baseline ERM method by more than 1 percentage point. They argue that DG methods without a model selection strategy remain incomplete and should therefore be specified as part of the method. A similar recommendation was done by Saito et al. (2021) for domain adaptation.

PDA methods have been designed using target labels at test time to select the best models. Related work (Saito et al., 2021; You et al., 2019) on model selection strategies for domain adaptation claimed to select the best models without using target labels. However, a realistic empirical study of these strategies in PDA is still lacking. In this work, we conduct extensive experiments to study the impact of model selection strategies on the performance of partial domain adaptation methods. We evaluate 6 different PDA methods over 7 different model selection strategies, 4 of which do not use target labels, and 2 different datasets under the same experimental protocol for a fair comparison. We list below our major findings:

- For a given method, the difference of accuracy attained by a model selected without target labels and a model selected with target labels can reach 30 percentage points (See Table 1 for a summary of results).

- Only 1 pair of PDA methods and target label-free model selection strategies achieve comparable accuracies to when target labels are used, while still improving over a source only baseline.

- Random seed plays an important role in the selection of hyper-parameters. Selected parameters are not stable across different seeds and the standard deviation between accuracies on the same task can be up to 8.4% even when relying on target labels for model selection.

- Under a more realistic scenario where some target labels are available, 100 random samples is enough to see only a drop of 1 percentage point in accuracy (when compared to using all target samples). However, using only one labeled target sample per class leads to a significant drop in performance.

**Related work.** Concurrent work Musgrave et al. (2022) also conducted a study of model selection methods for unsupervised domain adaptation. Similarly to our work, the authors study several model selection techniques and several domain adaptation methods, but their focus is in the standard domain adaptation setting, while we focus on PDA. In Section 4, we discuss our slightly different methodologies and in Section 5, we compare our respective findings. Other domain adaptation variants could have been considered like the open-set domain adaptation (Busto & Gall, 2017). However, these other variants require specific methods and it would have been computationally expensive to make an extensive study of each variant in a single work. We thus leave other domain adaptation variants for future work.

**Outline.** In Section 2, we provide an overview and a mathematical description of the different model selection strategies considered in this work. Then in Section 3, we discuss the PDA methods that we consider in this work. In Section 4 we describe the training procedures, hyper-parameter tuning and evaluation protocols

| DATASET | Model Selection | S. ONLY | PADA | SAFN | BA3US | AR | JUMBOT | MPOT |
|---|---|---|---|---|---|---|---|---|
| OFFICE-HOME | Worst (w/o target labels) | 59.55 (-2.31) | 52.72 (-11.00) | 61.37 (-1.93) | 62.25 (-13.73) | 64.32 (-8.42) | 61.28 (-15.87) | 46.92 (-30.38) |
| | Best (w/o target labels) | 60.73 (-1.14) | 63.08 (-0.64) | 62.59 (-0.71) | 75.37 (-0.61) | 70.58 (-2.16) | 74.61 (-2.54) | 66.24 (-11.07) |
| | Avg (w/o target labels) | $60.22 \pm 0.43$ | $59.5 \pm 4.1$ | $62.02 \pm 0.4$ | $69.9 \pm 5.1$ | $67.7 \pm 2.8$ | $67.8 \pm 5.8$ | $59.7 \pm 7.6$ |
| | ORACLE | 61.87 | 63.72 | 63.30 | 75.98 | 72.73 | 77.15 | 77.31 |
| VISDA | Worst (w/o target labels) | 55.02 (-4.46) | 32.32 (-22.26) | 42.83 (-19.81) | 51.07 (-16.60) | 55.69 (-18.15) | 59.86 (-24.15) | 61.62 (-25.33) |
| | Best (w/o target labels) | 55.24 (-4.24) | 56.83 (2.26) | 58.62 (-4.02) | 65.58 (-2.09) | 67.20 (-6.65) | 77.69 (-6.31) | 78.40 (-8.54) |
| | Avg (w/o target labels) | $55.1 \pm 0.1$ | $45.1 \pm 8.8$ | $51.1 \pm 7.4$ | $57.5 \pm 5.3$ | $63.5 \pm 4.6$ | $65.2 \pm 7.3$ | $71.2 \pm 6.26$ |
| | ORACLE | 59.48 | 54.57 | 62.64 | 67.67 | 73.85 | 84.01 | 86.95 |

Table 1: Task accuracy average computed over three different seeds (2020, 2021, 2022) on Partial OFFICE-HOME and Partial-VISDA. For each dataset and PDA method, we display the results of the *worst and best performing model selection that do not use target labels* as well as their average and the ORACLE model selection strategy. All results can be found in Table 6.

used to evaluate all methods fairly. In Section 5, we discuss the results of the different benchmarked methods and the performance of the different model selection strategies. Finally in Section 6, we give some recommendations for future work in partial domain adaptation.

## 2 Model Selection Strategies: An Overview

In UDA, the goal is to classify unlabeled data from a target domain leveraging labeled data from a source domain (Pan & Yang, 2010). Formally, let $\mathcal{D}_s$ (resp. $\mathcal{D}_t$) be the labeled (resp. unlabeled) source (resp. target) dataset which is composed of $n_s$ (resp. $n_t$) *i.i.d* random labeled (resp. unlabeled) vectors in $\mathbb{R}^d$ drawn from a distribution $\mathsf{p}$ (resp. $\mathsf{q}$), *i.e.,* $\mathcal{D}_s = \left\{ (\mathbf{x}_i^s, \mathbf{y}_i^s) \right\}_{i=1}^{n_s}, \mathbf{x}_i^s \in \mathbb{R}^d$ (resp. $\mathcal{D}_t = \left\{ (\mathbf{x}_j^t) \right\}_{j=1}^{n_t}, \mathbf{x}_j^t \in \mathbb{R}^d$). The goal is to find $f_\theta$, typically a deep neural network, that minimizes the target risk $\epsilon_t(f) = \mathbb{P}_{(\mathbf{x},y)\sim\mathsf{q}}[f(x) \neq y]$. In the standard domain adaptation, both domains share the same label space $\mathcal{Y}_s = \mathcal{Y}_t$, while in PDA the target classes form a subset of the source classes $\mathcal{Y}_t \subset \mathcal{Y}_s$. To classify the samples from the two domains, we rely on a feature extractor and a classifier. Formally, let $g_\theta : \mathcal{X} \mapsto \mathbb{R}^d$ be a feature extractor and $f_\phi : \mathbb{R}^d \mapsto \Sigma^C$ be a classifier, where $d$ is the dimension of feature space, $\Sigma$ is the simplex and $C$ be the number of classes. The maps $g_\theta$ and $f_\phi$ are usually neural networks parametrized by $\theta$ and $\phi$.

As discussed in introduction, model selection (choosing hyper-parameters, training checkpoints, neural network architectures) is a crucial part of training neural networks. In the supervised learning setting, a validation set is used to estimate the model's accuracy. However, in UDA such approach is not possible as we have unlabeled target samples. Several strategies have been designed to address this issue. Below, we discuss the ones used in this work.

**Source Accuracy** (S-ACC). Ganin & Lempitsky (2015) used the accuracy estimated on a small validation set from the source domain to perform the model selection. While the source and target accuracies are related, there are no theoretical guarantees. You et al. (2019) showed that when the domain gap is large this approach fails to select competitive models.

**Deep Embedded Validation** (DEV). Sugiyama et al. (2007) and Long et al. (2018) perform model selection through Importance-Weighted Cross-Validation (IWCV). Under the assumption that the source and target domain follow a covariate shift, the target risk can be estimated from the source risk through importance weights that give increased importance to source samples that are closer to target samples. Formally, the importance weights $w(\mathbf{x})$ correspond to the ratio of the target and source densities, i.e., $w(\mathbf{x}) = \frac{\mathsf{q}(\mathbf{x})}{\mathsf{p}(\mathbf{x})}$ and are found by estimating each distribution using Gaussian kernels. Recently, You et al. (2019) proposed an improved variant, Deep Embedded Validation (DEV), that controls the variance of the estimator, through a method called control variate, and estimates the importance weights with a discriminative model that distinguish source samples from target samples leading to a more stable and effective method. The latter is based on the fact that

$$\psi(\mathbf{x}) = \frac{1}{1 + w(\mathbf{x})}$$

where $\psi$ is an optimal source-target discriminator, meaning that if $\mathbf{x}$ is drawn from $\mathtt{p}$, $\psi(\mathbf{x}) = 1$ and $\psi(\mathbf{x}) = 0$ if $\mathbf{x}$ is drawn from $\mathtt{q}$.

**Entropy** (ENT). While minimizing the entropy of the target samples has been used in domain adaptation to improve accuracy by promoting tighter clusters, Morerio et al. (2018) showed that it can also be used for model selection. Formally, we want the classifier $f_{\phi^\star}$ and feature extractor $g_{\theta^\star}$ that have the lowest entropy *i.e.,* $\sum_{i=1}^{n_t} H\Big(f_\phi\big(g_\theta(\mathbf{x}_i^t)\big)\Big)$, where $H$ denotes the entropy. The intuition is that a lower entropy model corresponds to a highly confident model with discriminative target features and therefore reliable predictions.

**Soft Neighborhood Density** (SND). Saito et al. (2021) argue that a good UDA model will have a cluster structure where nearby target samples are in the same class. They claim that entropy is not able to capture this property and propose the Soft Neighborhood Density (SND) score to address it. Formally, they compute the similarity between extracted features from the target data $S_{i,j} = \langle g_\theta(\mathbf{x}_i^t), g_\theta(\mathbf{x}_j^t)\rangle$. Then, the SND score is defined as the entropy of the following probability distribution $P_{i,j} = \big(\exp(S_{i,j})/\tau\big)/\big(\sum_j \exp(S_{i,j})/\tau\big)$. The selected model $g_\theta$ is the model with the highest SND score.

**Target Accuracy** (ORACLE). We consider as well the target accuracy on all target samples. While we emphasize once again its use is not realistic in unsupervised domain adaptation (hence why we will refer to it as ORACLE), it has nonetheless been used to report the best accuracy achieved by the model along training in several previous works (Cao et al., 2018; Xu et al., 2019; Jian et al., 2020; Gu et al., 2021; Nguyen et al., 2022). Here, we use it as an upper bound for all the other model selection strategies and to check the reproducibility of previous works.

**Small Labeled Target Set** (1-SHOT and 100-RND). For real-world applications in an industry setting, it is unlikely that a model will be deployed without an estimation of its performance on the target domain. Therefore, one can imagine a situation where a PDA method is used and a small set of target samples is available. Thus, we will compute the target accuracy with 1 labeled sample per class (1-SHOT) and 100 random labeled target samples (100-RND) as model selection strategies. One could argue that the 100 random samples could have been used in the training with semi-supervised domain adaptation methods. However, note that we do not know how many classes we have on the target domain so it is hard to form a split when we have uncertainty of classes. For instance, 100-RND represents possibly less than 2 samples per class for one of our real-world dataset, as we do not know the number of classes, making a potential split between a train and validation target sets not possible.

## 3 Partial Domain Adaptation Methods

In this section, we give a brief description of the PDA methods considered in our study. They can be grouped into two families: adversarial training and divergence minimization.

**Adversarial training.** To solve the UDA problem, Ganin et al. (2016) aligned the source and target domains with the help of a domain discriminator trained adversarially to be able to distinguish the samples from the two domains. In addition to $g_\theta$ and $f_\phi$, we also consider an adversarial classifier $D_\zeta : \mathcal{Z} \mapsto \{0,1\}$ parameterized by $\zeta$. The maps are trained in a adversarial manner where $D_\zeta$ tries to distinguish the source data from the target data, while $g_\theta$ is trained to confuse the domain discriminator $D_\zeta$:

$$\zeta^\star, \theta^\star = \arg\min_\theta \arg\max_\zeta \frac{1}{n_s}\sum_{i=1}^{n_s} \log\Big(D_\zeta\big(g_\theta(\mathbf{x}_i^s)\big)\Big) + \frac{1}{n_t}\sum_{j=1}^{n_t} \log\Big(1 - D_\zeta\big(g_\theta(\mathbf{x}_j^t)\big)\Big).$$

However, when naively applied to the PDA setting, this strategy aligns target samples to source sample whose classes are not in the target domain and as a result the model performs worse than a model trained only on source data. To remedy this problem, Cao et al. (2018) proposed PADA that introduces a PDA specific solution to adversarial domain adaptation: the contribution of the source-only class samples to the training of both the source classifier and the domain adversarial network is decreased. This is achieved through class weights that are calculated by simply averaging the classifier prediction on all target samples. As the

| PDA Methods | PADA, SAFN, BA3US, AR, JUMBOT, MPOT |
|---|---|
| Model Selection Strategies | S-ACC, ENT, DEV, SND, 1-SHOT, 100-RND, ORACLE |
| Architecture | ResNet50 backbone ⊕ linear bottleneck ⊕ linear classification head |
| Experimental protocol | 3 seeds on the 12 tasks of OFFICE-HOME and 2 tasks of VISDA |

Table 2: Summary of our considered methods, model selection strategies, architecture and datasets, where ⊕ denotes the operation of adding layers.

| Method | Architecture (bottleneck) | Runs per tasks | Model Selection Hyper-Parameters | Model Selection Along Training |
|---|---|---|---|---|
| PADA | Linear | 1 | IWCV (lacks details) | ORACLE |
| SAFN | Non-Linear | 3 | Unknown | ORACLE |
| BA3US | Linear | 3 | Unknown | ORACLE |
| AR | Non-Linear | 1 | IWCV (lacks details) | ORACLE |
| JUMBOT | Linear | 1 | ORACLE | FINAL |
| MPOT | Linear | 3 | Unknown | ORACLE |

Table 3: Summary of the experimental protocol used for SOTA partial domain adaptation methods. We refer to Appendix A.1 for additional details.

source-only classes should not be predicted in the target domain, they should have lower weights. More recently, Jian et al. (2020) proposed BA3US which augments the target mini-batch with source samples to transform the PDA problem into a vanilla DA problem. In addition, an adaptive weighted entropy objective is used to encourage incorrect classes to have uniform and low prediction scores.

**Divergence minimization.** Another standard direction to align the source and target distributions in the feature space of a neural network is to minimize a given divergence between distributions of domains. The purpose of the minimization divergence approach is to find the optimal parameters $\theta^\star$ which minimizes the divergence $L$:

$$\theta^\star = \mathrm{argmin}_\theta \, L\big(g_{\theta\#}\mathtt{p}, g_{\theta\#}\mathtt{q}\big),$$

where $g_{\theta\#}$ denotes the pushforward operator of the map $g_\theta$. Xu et al. (2019) empirically found that target samples have low feature norm compared to source samples. Based on this insight, they proposed SAFN which progressively adapts the feature norms of the two domains by minimizing the Maximum Mean Feature Norm Discrepancy as the divergence $L$ (Gretton et al., 2012). Other approaches consider the optimal transport (OT) cost as the divergence $L$ (Bhushan Damodaran et al., 2018) with mini-batches (Peyré & Cuturi, 2019; Fatras et al., 2020; 2021b). For the PDA problem in specific, (Fatras et al., 2021a) developed JUMBOT, a mini-batch unbalanced optimal transport that learns a joint distribution of the embedded samples and labels. The use of unbalanced OT is critical for the PDA problem as it allows to transport only the source samples whose classes are in the target domain. Based on this work, (Nguyen et al., 2022) investigated the partial OT variant (Chapel et al., 2020), a particular case of unbalanced OT, proposing M-POT. Finally, another line of work is to use the Kantorovich-Rubenstein duality of optimal transport to perform the alignment similarly to WGAN (Arjovsky et al., 2017). This is precisely the work of Gu et al. (2021) that proposed, AR. In addition, source samples are reweighted in order to not transport the source-only class samples. The Kantorovich-Rubenstein duality relies on a one Lipschitz function which is approximated using adversarial training like the PDA methods described above.

## 4 Experimental Protocol

In this section, we discuss our choices regarding the training details, datasets and neural network architecture. We then discuss the hyper-parameter tuning used in this work. We summarize the PDA methods, model selection strategies and experimental protocol used in this work in Table 2. The main differences in the experimental protocol of the different published state-of-the-art (SOTA) methods is summarized in Table 3.

| METHOD | S. ONLY | PADA | SAFN | BA3US | AR | JUMBOT | MPOT |
|---|---|---|---|---|---|---|---|
| Reported | 61.35 | 62.06 | 71.84 | 75.98 | 77.11 | 75.47 | 77.98 |
| Reimplementation | 61.87 | 63.72 | 74.72 | 75.98 | 76.00 | 77.15 | 77.31 |
| Ours | 61.87 | 63.72 | 63.30 | 75.98 | 72.73 | 77.15 | 77.31 |

Table 4: Comparison between reported accuracies on partial OFFICE-HOME from published methods with our reimplementation using the ORACLE model selection strategy. Ours denotes our reimplementation where all methods have the same bottleneck architectures as discussed in Section 4.

To perform our experiments we developed a PyTorch (Paszke et al., 2019) framework: BenchmarkPDA. We make it available for researchers to use and contribute with new algorithms and model selection strategies:



https://github.com/oberman-lab/BenchmarkPDA



It is the standard in the literature when proposing a new method to report directly the results of its competitors from the original papers (Cao et al., 2018; Xu et al., 2019; Jian et al., 2020; Gu et al., 2021; Nguyen et al., 2022). As a result some methods differ for instance in the neural network architecture implementation (AR (Gu et al., 2021), SAFN (Xu et al., 2019)) or evaluation protocol JUMBOT (Fatras et al., 2021a) with other methods. These changes often contribute to an increased performance of the newly proposed method leaving previous methods at a disadvantage. Therefore we chose to implement all methods with the same commonly used neural network architecture, optimizer, learning rate schedule and evaluation protocol. We discuss the details below.

## 4.1 Methods, Datasets, Training and Evaluation Details

**Methods.** We implemented 6 PDA methods by adapting the code from the Official GitHub repositories of each method: Source Only, PADA (Cao et al., 2018), SAFN (Xu et al., 2019), BA3US (Jian et al., 2020), AR (Gu et al., 2021), JUMBOT (Fatras et al., 2021a), MPOT (Nguyen et al., 2022). We provide the links to the different official repositories in Appendix A.1. A comparison with previous reported results can be found in Table 11 and we postpone the discussion to Section 5.

**Datasets.** We consider two standard real-world datasets used in DA. Our first dataset is OFFICE-HOME (Venkateswara et al., 2017). It is a difficult dataset for unsupervised domain adaptation (UDA), it has 15,500 images from four different domains: Art (A), Clipart (C), Product (P) and Real-World (R). For each domain, the dataset contains images of 65 object categories that are common in office and home scenarios. For the partial OFFICE-HOME setting, we follow Cao et al. (2018) and select the first 25 categories (in alphabetic order) in each domain as a partial target domain. We evaluate all methods on all 12 possible tasks, where by task we mean training a classifier using one domain as source data and another different domain as target data. For example, the AC task is the partial domain adaptation scenario where the Art domain is used as source and the Clipart domain is used as target. VISDA (Peng et al., 2017) is a large-scale dataset for UDA. It has 152,397 synthetic images and 55,388 real-world images, where 12 object categories are shared by these two domains. For the partial VISDA setting, we follow Cao et al. (2018) and select the first 6 categories, taken in alphabetic order, in each domain as a partial target domain. We evaluate the models in the two possible scenarios. We highlight that we are the first to investigate the performance of JUMBOT and MPOT on partial VISDA.

**Model Selection Strategies** We consider the 7 different strategies for model selection described in Section 2: S-ACC, DEV, ENT, SND, ORACLE, 1-SHOT, 100-RND. We use them both for hyper-parameter tuning as well selecting the best model along training. Since S-ACC, DEV and SND require a source validation set, we divide the source samples into a training subset (80%) and validation subset (20%). Regardless of the model selection strategy used, all methods are trained using the source training subset. This is in contrast with previous work that uses all source samples, but necessary to ensure a fair comparison of the model selection strategies. We refer to Appendix A.2 for additional details.

| Dataset | Variant | BA3US | | | JUMBOT | | | MPOT | | | SAFN | | |
|---|---|---|---|---|---|---|---|---|---|---|---|---|---|
| | | ENT | DEV | SND | ENT | DEV | SND | ENT | DEV | SND | ENT | DEV | SND |
| OFFICE-HOME | Naive | 52.60 | **63.10** | 44.48 | 52.30 | 26.75 | 17.67 | **49.01** | 16.72 | **30.63** | 32.12 | **49.67** | 5.01 |
| | Heuristic | **58.45** | **63.10** | **60.96** | **56.24** | **45.79** | **55.16** | **49.01** | **45.61** | **30.63** | 46.27 | **49.67** | **49.67** |
| VISDA | Naive | 39.06 | **36.99** | 1.14 | 35.89 | **54.53** | 11.99 | **75.04** | **55.33** | 36.11 | **52.82** | **53.26** | 0.83 |
| | Heuristic | **67.50** | 34.94 | **38.76** | **47.23** | **54.53** | **66.42** | **75.04** | **55.33** | **85.36** | **52.82** | **53.26** | **52.82** |

Table 5: Comparison between the naive model selection strategy and our heuristic approach. Accuracy on AC task for OFFICE-HOME and SR task for VISDA. Best results in **bold**.

**Architecture.** Our network is composed of a feature extractor with a linear classification layer on top of it. The feature extractor is a ResNet50 (He et al., 2016), pre-trained on ImageNet (Deng et al., 2009), with its last linear layer removed and replaced by a linear bottleneck layer of dimension 256. This architecture is used by almost all methods with the exception of SAFN and AR which use different architectures. However we believe that it is important to use the same architecture for all methods to understand their own performance. While it is possible that we underestimate the performance of SAFN and AR as a result, it is also possible that other methods benefit from these different architectures. We leave this question for future work.

**Optimizer.** We use the SGD (Robbins & Monro, 1951) algorithm with momentum of 0.9, a weight decay of $5e^{-4}$ and Nesterov acceleration. As the bottleneck and classifier layers are randomly initialized, we set their learning rates to be 10 times that of the pre-trained ResNet50 backbone. We schedule the learning rate with a strategy similar to the one in (Ganin et al., 2016): $\chi_p = \frac{\chi_0}{(1+\mu i)^{-\nu}}$, where $i$ is the current iteration, $\chi_0 = 0.001$, $\gamma = 0.001$, $\nu = 0.75$. While this schedule is slightly different than the one reported in previous work, it is the one implemented in the different official code implementations. We elaborate in the Appendix A.3 on the differences and provide additional details. Finally, as for the mini-batch size, JUMBOT and M-POT were designed with a stratified sampling, *i.e.,* a balanced source mini-batch with the same number of samples per class. This strategy reduces the optimal transport matching between target samples and source samples from source-only classes, which in turn leads to less target samples being classified as belonging to source-only classes. On the other hand, it was shown that for some methods (e.g. BA3US) using a larger mini-batch, than what was reported, leads to a decreased performance (Fatras et al., 2021a). As a result, we used the default mini-batch strategies for each method. JUMBOT and M-POT use stratified mini-batches of size 65 for OFFICE-HOME and 36 for VISDA. All other methods use a random uniform sampling strategy with a mini-batch size of 36.

**Evaluation Protocol.** For the hyper-parameters chosen with each model selection strategy, we run the methods for each tasks 3 times, each with a different seed (2020, 2021, 2022). We tried to control for the randomness across methods by setting the seeds at the beginning of training. Interestingly, as we discuss in more detail in Section 5, some methods demonstrated a non-negligible variance across the different seeds showing that some hyper-parameters and methods are not robust to randomness.

### 4.2 Hyper-Parameter Tuning

Previous works (Gulrajani & Lopez-Paz, 2021; Musgrave et al., 2021; 2022) perform random searches with the same number of runs for each method. In contrast, we perform hyper-parameter grid searches for each method. As a result, the hyper-parameter tuning budgets differs across the methods depending on the number of hyper-parameters and the chosen grid. While one can argue this leads to an unfair comparison of the methods, in practice in most real-world applications one will be interested in using the best method that our approach will precisely capture.

The hyper-parameter tuning needs to be performed for each tasks of each dataset, but that would require a significant computational resources without a clear added benefit. Instead for each dataset, we first perform the hyper-parameter tuning on trained models over a single task: A2C for OFFICE-HOME and S2R for VISDA. Then in a second time, we use the selected hyper-parameters from the first phase on the remaining tasks. This same strategy was adopted in (Fatras et al., 2021a) and the hyper-parameters were found to generalize to the remaining tasks in the dataset. We conjecture that this may be due to the fact that information

regarding the number of target only classes is implicitly hidden in the hyper-parameters. See Appendix A.4 for more details regarding the hyper-parameters.

In practice during the first phase, we set the number of training iterations to 5000 steps for OFFICE-HOME and 10000 steps for VISDA. Then, we apply the different model selection strategies to the different trained models and select the hyper-parameters that optimize the model selection strategies. As we are looking for the best hyper-parameters on a single task that hopefully generalize to the remaining tasks, we do not consider the end of training as a hyper-parameter during this phase. In the second phase, we use the chosen hyper-parameters and run the methods on all tasks. We use the model selection strategies to select the best model for each task along training, effectively considering the 'end of training' as a hyper-parameter.

With the wrong choice of hyper-parameters, we can end up with a degenerate model that predicts the same label for all examples with high confidence. This model will have good DEV, SND and ENT scores (being highly confident means it has low entropy) and therefore is considered a 'good' model according to these criterias when in fact it's quite the contrary.

Several runs in our hyper-parameter search for JUMBOT, M-POT and BA3US were unsuccessful with the optimization reaching its end without the model being trained at all. The extreme case is a degenerate model that predicts the same label for all examples with high confidence. This poses a challenge to DEV, SND and ENT. A highly confident model has low entropy and therefore is considered a 'good' model according to ENT. A similar reasoning can be made for DEV and SND. This failure mode is accounted for in (Saito et al., 2021). Following their recommendations, for JUMBOT, M-POT and BA3US, before applying the model selection strategy, we discard models whose source domain accuracy is below a certain threshold $thr$, which is set with the heuristic as $thr = 0.9 \cdot Acc$. Here Acc denotes the source domain accuracy of the Source-Only model. In our experiments, this leads to select models whom the source accuracy is at least of $thr = 69.01\%$ for the A2C tasks on OFFICE-HOME and $thr = 89.83\%$ for the S2R tasks on VISDA. We choose this heuristic because the ablation study of some methods showed that doing the adaptation decreased slightly the source accuracy (Bhushan Damodaran et al., 2018). Table 5 shows that our heuristic leads to improved results.

Lastly, when choosing the hyper-parameters, we only consider the model at the end of training, discarding the intermediate checkpoint models in order to select hyper-parameters which do not lead to overfitting at the end of training and better generalize to the other tasks. Following the above protocol, for each dataset we trained *468* models in total in order to find the best hyper-parameters. Then, to obtain the results with our neural network architecture on all tasks of each dataset, we trained an additional *1224* models for OFFICE-HOME and *156* models for VISDA. We additionally trained 231 models with the different neural network architectures for AR and SAFN. In total, *2547* models were trained to make this study and we present the different results in the next section.

## 5 Partial domain adaptation experiments

We start the results section by discussing the differences between our reproduced results and the published results from the different PDA methods. Then, we compare the performance of the different model selection strategies. Finally, we discuss the sensitivity of methods to the random seed.

### 5.1 Reproducibility Of Previous Results

We start by ensuring that our reimplementation of PDA methods was done correctly by comparing our reproduced results with previously reported results in Table 11. As such the model selection strategy used is ORACLE. On OFFICE-HOME, both PADA and JUMBOT achieved higher average tasks accuracy (1.6 and 1.7 percentage points, respectively) in our reimplementation, while for BA3US and MPOT we recover the reported accuracy in their respective papers. However, we saw a decrease in performance for both SAFN and AR of roughly 8 and 5 percentage points respectively. This is to be expected due to the differences in the neural network architectures. While we use a linear bottleneck layer, SAFN uses a nonlinear bottleneck layer. As for AR, they make two significant changes: the linear classification head is replaced by a spherical logistic regression (SLR) layer (Gu et al., 2020) and the features are normalized (the 2-norm is set to a dataset dependent value, another hyper-parameter that requires tuning) before feeding them to the classification

| Dataset | Method | S-ACC | ENT | DEV | SND | 1-SHOT | 100-RND | ORACLE |
|---|---|---|---|---|---|---|---|---|
| OFFICE-HOME | S. ONLY | 60.38±0.5 | 60.73±0.2 | 60.22±0.3 | 59.55±0.3 | 58.92±0.4 | 60.34±0.4 | 61.87±0.3 |
| | PADA | 63.08±0.3 | 59.74±0.5 | 52.72±2.8 | 62.36±0.4 | 62.00±0.5 | 63.22±0.1 | 63.72±0.3 |
| | SAFN | 62.09±0.2 | 61.37±0.3 | 62.03±0.4 | 62.59±0.1 | 49.30±0.7 | 62.36±0.2 | 63.30±0.2 |
| | BA3US | 68.32±1.1 | 73.36±0.6 | 62.25±7.1 | 75.37±0.8 | 65.56±7.6 | 75.19±0.4 | 75.98±0.3 |
| | AR | 65.68±0.3 | 70.58±0.4 | 64.32±0.9 | 70.25±0.2 | 70.56±0.7 | 70.34±0.2 | 72.73±0.3 |
| | JUMBOT | 62.89±0.2 | 74.61±0.8 | 61.28±0.1 | 72.29±0.2 | 74.95±0.1 | 75.74±0.3 | 77.15±0.4 |
| | MPOT | 66.24±0.1 | 64.46±0.1 | 61.37±0.2 | 46.92±0.4 | 68.28±0.2 | 73.06±0.3 | 77.31±0.5 |
| VISDA | S. ONLY | 55.15±2.4 | 55.24±3.2 | 55.07±1.2 | 55.02±2.9 | 55.72±2.2 | 58.16±0.6 | 59.48±0.4 |
| | PADA | 47.48±4.8 | 32.32±4.9 | 43.43±5.3 | 56.83±1.0 | 53.15±2.9 | 54.38±2.7 | 54.57±2.6 |
| | SAFN | 58.20±1.7 | 42.83±6.3 | 58.62±1.3 | 44.82±8.8 | 56.89±2.1 | 59.09±2.8 | 62.64±1.5 |
| | BA3US | 55.10±3.7 | 65.58±1.4 | 58.40±1.4 | 51.07±4.3 | 64.77±1.4 | 67.44±1.2 | 67.67±1.3 |
| | AR | 66.68±1.0 | 64.27±3.6 | 67.20±1.5 | 55.69±0.9 | 70.29±1.7 | 72.60±0.8 | 73.85±0.9 |
| | JUMBOT | 60.63±0.7 | 62.42±2.4 | 59.86±0.6 | 77.69±4.2 | 78.34±1.9 | 83.49±1.9 | 84.01±1.9 |
| | MPOT | 70.02±2.0 | 74.64±4.4 | 61.62±1.3 | 78.40±3.9 | 70.96±3.7 | 86.69±5.1 | 86.95±5.0 |

Table 6: Task accuracy average over seeds 2020, 2021, 2022 on Partial OFFICE-HOME and Partial VISDA for the PDA methods and model selection strategy. For each method, we highlight the best and worst label-free model selection strategies in green and red, respectively.

head. While we account for the first change by comparing to AR (w/ linear) results reported in (Gu et al., 2021), in our neural network architecture we do not normalize the features. These changes, nonlinear bottleneck layer for SAFN and feature normalization for AR, significantly boost the performance of both methods. Performing an ablation study of new architectures with respect to the original one, as done in (Gu et al., 2021), allows us to understand the architecture's influence on the method. We recommend authors to perform one as it also makes the method easier to reproduce results. When now comparing our reimplementation with the same neural network architectures, our SAFN reimplementation achieves a higher average tasks accuracy by 3 percentage points, while our AR reimplementation is now only 1 percentage points below. The fact that AR reported results are from only one run, while ours are averaged across 3 distinct seeds, justifies the small remaining gap. Moreover, we report higher accuracy or on par on 4 of the 12 tasks. Given all the above and further discussion of the VISDA dataset results in Appendix B, our reimplementations are trustworthy and give validity to the results we discuss in the next sections.

## 5.2 Results for Model Selection Strategies

**Model Selection Strategies (w/ vs w/o target labels)** All average accuracies on the OFFICE-HOME and VISDA datasets can be found in Table 6. For all methods on OFFICE-HOME, we can see that the results for model selection strategies which do not use target labels are below the results given by ORACLE. For some pairs, the drop of performance can be significant, leading some methods to perform on par with the S. ONLY method. That is the case on OFFICE-HOME when DEV is paired with either BA3US, JUMBOT and MPOT. Even worse is MPOT with SND as the average accuracy is more than 10 percentage points below that of S. ONLY with any model selection strategy. Overall on OFFICE-HOME, except for MPOT, all methods when paired with either ENT or SND give results that are at most 2 percentage points below compared to when paired with ORACLE.

A similar situation can be seen over the VISDA dataset where the accuracy without target labels can be down to 25 percentage points. Yet again, some model selection strategies can lead to scores even worse than S. ONLY. That is the case for PADA, SAFN and BA3US. Contrary to OFFICE-HOME, all model selection strategies without target labels lead to at least one method with results on par or worse in comparison to the S. ONLY method. More generally, no model selection strategy without target labels can lead to score on par to the ORACLE model selection strategy. Finally, PADA performs worse than S. ONLY for most model selection strategies, including the ones which use target labels. Perhaps a little surprising, when combined with SND it achieved a higher per run average than with ORACLE, although within the standard deviation. This is also a consequence of the random seed dependence mentioned before on VISDA: as the hyper-parameters were

chosen by performing just one run, we were simply "unlucky". In general, all of this confirms the standard assumption in the literature regarding the difficulty of the VISDA dataset.

These experiments also allow us to draw some conclusions regarding the robustness of methods to model selection strategies with respect to the number of hyper-parameters. We see that PADA and BA3US are not robust for either one of the datasets, while SAFN is robust to the choice of model selection strategy for OFFICE-HOME. These are the methods with fewest hyper-parameters ruling out the fact that less hyper-parameters leads to more robust methods and suggesting it is method and dataset specific. Furthermore regarding reliability, we find that the optimal transport based approaches (MPOT, JUMBOT) to be the most sensitive to model selection strategies as they exhibit the largest performance gaps between the best and worse model selection strategies without target labels. It shows, that while they are able to achieve SOTA results with the ORACLE model selection strategy, that performance is highly tied to the hyper-parameter choice.

**Model Selection Strategies (w/ target labels)** We recall that the ORACLE model selection strategy uses all the target samples to compute the accuracy while 1-SHOT and 100-RND use only subsets: 1-SHOT has only one sample per class for a total of 25 and 6 on OFFICE-HOME and VISDA, respectively, while 100-RND has 100 random target samples. Our results show that using only 100 random target labeled samples is enough to reasonably approximate the target accuracy leading to only a small accuracy drop (one percentage point in almost all cases) for both datasets. Not surprisingly, the gap between the 1-SHOT and ORACLE model selection strategies is even bigger, leading in some instances to worse results than with a model selection strategy that uses no target labels. This poor performance of the 1-SHOT model selection strategy also highlights that semi-supervised domain adaptation (SSDA) methods are not a straightforward alternative to the 100-RND model selection strategy. While one could argue that the target labels could be leveraged during training like in SSDA methods, one still needs labeled target data to perform model selection. However our results suggest that we would need at least 3 samples per class for SSDA methods. In addition, knowing that we have a certain number of labeled samples per class provides information regarding which classes are target only, one of the main assumptions in PDA. In that case, PDA methods could be tweaked. This warrants further study that we leave as future work. Finally, we have also investigated a smaller labeled target set of 50 random samples (50-RND) instead of 100 random samples. The accuracies of methods using 50-RND were not as good as when using 100-RND. All results of pairs of methods and 50-RND can be found in Appendix B. The smaller performance show that the size of the labeled target set is an important element and we suggest to use at least 100 random samples.

**Model Selection Strategies (w/o target labels)** Among all 49 pairs of methods and model selection strategies, only the (BA3US, ENT) pair achieved average tasks accuracies within 3 percentage points of its ORACLE counterpart (*i.e.*, (BA3US, ORACLE)), while improving over S. ONLY model. Our experiments show that there is no model selection strategy which performs well for all methods. That is why to deploy models in a real-world scenario, we advise to test selected models on a small labeled target set (*i.e.,* 100-RND)) to assess the performance of the models as model selection strategies without target labels can perform poorly.

Our conclusion is that the model selection for PDA methods is still an open problem. We conjecture that it is also the case for domain adaptation as the considered metrics were developed first for this setting. For future proposed methods, researchers should specify not only which model selection strategy should be used, but also which hyper-parameter search grid should be considered, to deploy them in a real-world scenario.

**Comparison with other model selection strategy studies** Our study of model selection strategies is related to the Gulrajani & Lopez-Paz (2021); Saito et al. (2021); Musgrave et al. (2022). In their respective study, Gulrajani & Lopez-Paz (2021) for domain generalization and Saito et al. (2021); Musgrave et al. (2022) for unsupervised domain adaptation argue that the methods remain incomplete without model selection strategies as the latter can have a big impact on their performance. Our findings are similar to theirs and we recommend that for each new partial domain adaptation method, its authors should recommend a target label-free model selection strategy.

| Task | Method | s-acc | ent | dev | snd | 1-shot | 100-rnd | oracle |
|------|--------|-------|-----|-----|-----|--------|---------|--------|
| S2R | S. ONLY | $46.96 \pm 1.5$ | $48.17 \pm 3.9$ | $49.00 \pm 0.9$ | $48.17 \pm 3.9$ | $49.43 \pm 0.8$ | $50.01 \pm 1.6$ | $51.86 \pm 1.4$ |
| | PADA | $44.56 \pm 5.9$ | $40.83 \pm 11.3$ | $41.04 \pm 4.3$ | $56.14 \pm 9.7$ | $52.94 \pm 4.3$ | $49.34 \pm 8.4$ | $49.34 \pm 8.4$ |
| | SAFN | $52.04 \pm 3.5$ | $29.86 \pm 16.7$ | $52.42 \pm 2.9$ | $28.46 \pm 16.5$ | $49.97 \pm 3.3$ | $47.83 \pm 0.6$ | $56.88 \pm 2.1$ |
| | BA3US | $44.21 \pm 3.0$ | $71.17 \pm 1.9$ | $48.78 \pm 1.9$ | $46.12 \pm 7.8$ | $66.79 \pm 1.5$ | $71.45 \pm 0.8$ | $71.77 \pm 1.1$ |
| | AR | $68.39 \pm 1.3$ | $75.28 \pm 2.9$ | $68.54 \pm 1.3$ | $57.61 \pm 0.4$ | $70.11 \pm 1.4$ | $75.09 \pm 5.2$ | $76.33 \pm 4.5$ |
| | JUMBOT | $55.23 \pm 2.3$ | $56.25 \pm 2.1$ | $54.35 \pm 2.0$ | $75.23 \pm 8.4$ | $81.27 \pm 6.9$ | $89.94 \pm 1.1$ | $90.55 \pm 0.5$ |
| | MPOT | $64.57 \pm 2.9$ | $82.10 \pm 2.0$ | $57.02 \pm 1.5$ | $84.45 \pm 0.4$ | $71.33 \pm 4.4$ | $87.20 \pm 2.3$ | $87.23 \pm 2.3$ |
| R2S | S. ONLY | $63.34 \pm 3.4$ | $62.32 \pm 2.7$ | $61.13 \pm 3.3$ | $61.88 \pm 2.3$ | $62.00 \pm 3.9$ | $66.30 \pm 2.0$ | $67.11 \pm 2.1$ |
| | PADA | $50.39 \pm 3.8$ | $23.80 \pm 1.6$ | $45.82 \pm 9.2$ | $57.53 \pm 10.3$ | $53.36 \pm 1.7$ | $59.43 \pm 5.8$ | $59.81 \pm 6.2$ |
| | SAFN | $64.37 \pm 0.7$ | $55.80 \pm 5.2$ | $64.82 \pm 0.5$ | $61.19 \pm 3.3$ | $63.82 \pm 1.0$ | $70.34 \pm 5.8$ | $68.40 \pm 1.2$ |
| | BA3US | $65.99 \pm 4.6$ | $59.99 \pm 1.3$ | $68.01 \pm 1.9$ | $56.01 \pm 2.9$ | $62.75 \pm 2.6$ | $63.44 \pm 1.9$ | $63.56 \pm 1.8$ |
| | AR | $64.97 \pm 0.8$ | $53.26 \pm 9.7$ | $65.86 \pm 3.5$ | $53.78 \pm 2.1$ | $70.46 \pm 4.7$ | $70.11 \pm 5.0$ | $71.36 \pm 5.5$ |
| | JUMBOT | $66.04 \pm 1.0$ | $68.59 \pm 4.6$ | $65.36 \pm 0.8$ | $80.16 \pm 1.1$ | $75.42 \pm 4.8$ | $77.03 \pm 2.7$ | $77.46 \pm 3.3$ |
| | MPOT | $75.47 \pm 3.8$ | $67.18 \pm 9.1$ | $66.21 \pm 1.2$ | $72.36 \pm 7.4$ | $70.58 \pm 3.1$ | $86.18 \pm 8.1$ | $86.67 \pm 7.8$ |
| Avg | S. ONLY | $55.15 \pm 2.4$ | $55.24 \pm 3.2$ | $55.07 \pm 1.2$ | $55.02 \pm 2.9$ | $55.72 \pm 2.2$ | $58.16 \pm 0.6$ | $59.48 \pm 0.4$ |
| | PADA | $47.48 \pm 4.8$ | $32.32 \pm 4.9$ | $43.43 \pm 5.3$ | $56.83 \pm 1.0$ | $53.15 \pm 2.9$ | $54.38 \pm 2.7$ | $54.57 \pm 2.6$ |
| | SAFN | $58.20 \pm 1.7$ | $42.83 \pm 6.3$ | $58.62 \pm 1.3$ | $44.82 \pm 8.8$ | $56.89 \pm 2.1$ | $59.09 \pm 2.8$ | $62.64 \pm 1.5$ |
| | BA3US | $55.10 \pm 3.7$ | $65.58 \pm 1.4$ | $58.40 \pm 1.4$ | $51.07 \pm 4.3$ | $64.77 \pm 1.4$ | $67.44 \pm 1.2$ | $67.67 \pm 1.3$ |
| | AR | $66.68 \pm 1.0$ | $64.27 \pm 3.6$ | $67.20 \pm 1.5$ | $55.69 \pm 0.9$ | $70.29 \pm 1.7$ | $72.60 \pm 0.8$ | $73.85 \pm 0.9$ |
| | JUMBOT | $60.63 \pm 0.7$ | $62.42 \pm 2.4$ | $59.86 \pm 0.6$ | $77.69 \pm 4.2$ | $78.34 \pm 1.9$ | $83.49 \pm 1.9$ | $84.01 \pm 1.9$ |
| | MPOT | $70.02 \pm 2.0$ | $74.64 \pm 4.4$ | $61.62 \pm 1.3$ | $78.40 \pm 3.9$ | $70.96 \pm 3.7$ | $86.69 \pm 5.1$ | $86.95 \pm 5.0$ |

Table 7: Accuracy of different PDA methods based on different model selection strategies on the 2 Partial VISDA tasks. Average is done over three seeds (2020, 2021, 2022). For each method, we highlight the best and worst label-free model selection strategies in green and red, respectively.

## 5.3 Random Seed Dependence

Ideally, PDA methods should be robust to the choice of random seed. This is of particular importance when performing hyper-parameter tuning since typically only one run per set of hyper-parameters is done (that was the case in our work as well). We investigate this robustness by averaging all the results presented over three different seeds (2020, 2021 and 2022) and reporting the standard deviations. This is in contrast with previous work where only a single run is reported (Fatras et al., 2021a; Gu et al., 2021). Other works (Cao et al., 2018; Xu et al., 2019; Jian et al., 2020) that report standard deviations do not specify if the random seed is different across runs. Results for all tasks on VISDA dataset are in Table 7 and on OFFICE-HOME in Appendix B due to space constraints.

Our experiments show that some methods express a non-negligible instabilities over randomness with respect to any model selection methods. This is particularly true for BA3US when paired with DEV and 1-SHOT as model selection strategies: there are several tasks where the standard deviation is above 10%. While in this case this instability may stem from the poor performance of the model selection strategies, it is also visible when ORACLE is the model selection strategy used. For instance, the M-POT has a standard deviation of 3.3% on the AP tasks of OFFICE-HOME which corresponds to a variance of 11%. On VISDA this instability and seed dependence is even larger.

## 6 Conclusion

In this paper, we investigated how model selection strategies affect the performance of PDA methods. We performed a quantitative study with six PDA methods and seven model selection strategies on two real-word datasets. Based on our findings, we provide the following recommendations:

*i) Target label samples should be used to test models before using them in real-world scenario.* While this breaks the main PDA assumption, it is impossible to confidently deploy PDA models selected without the

use of target labels. Indeed, model selection strategies without target labels lead to a significant drop in performance in most cases in comparison to using a small validation set. We argue that the cost of labeling it outweighs the uncertainty in current model selection strategies.

*ii) The robustness of new PDA method to randomness should be tested over at least three different seeds.* We suggest to use the seeds (2020, 2021, 2022) to allow for a fair comparison with our results.

*iii) An ablation study should be considered when a novel architecture is proposed to quantify the associated increase of performance.*

As our work focus on a quantitative study of model selection methods and reproducibility of state-of-the-art partial domain adaptation methods, we do not see any potential ethical concern. Future work will investigate new model selection strategies which can achieve similar results as model selection strategies which use label target samples.

### Acknowledgments

This work was partially supported by NSERC Discovery grant (RGPIN-2019-06512) and a Samsung grant. Thanks also to CIFAR for their support through the Canada CIFAR AI Chairs program. Authors thank Christos Tsirigotis and Chen Sun for early comments on the manuscript.

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

# A Reproducible and Realistic Evaluation of Partial Domain Adaptation Methods

# Supplementary material

**Outline.** The supplementary material of this paper is organized as follows:

- In Section A, we give more details on our experimental protocol.

- In Section B, we provide additional results from our experiments.

## A Additional details on Experimental Protocol

### A.1 Implementations in BenchmarkPDA

In order to reimplement the different PDA methods, we adapted the code from the official repository associated with each of the paper. We list them in Table 8.

| Method | Code Repository |
|--------|-----------------|
| PADA | `https://github.com/thuml/PADA/blob/master/pytorch/src/` |
| SAFN | `https://github.com/jihanyang/AFN/blob/master/partial/OfficeHome/SAFN/code/` |
| BA3US | `https://github.com/tim-learn/BA3US/` |
| AR | `https://github.com/XJTU-XGU/Adversarial-Reweighting-for-Partial-Domain-Adaptation` |
| JUMBOT | `https://github.com/kilianFatras/JUMBOT` |
| M-POT | `https://github.com/UT-Austin-Data-Science-Group/Mini-batch-OT/tree/master/PartialDA` |

Table 8: Office Github code repositories for the PDA methods considered in this work.

One of our main claims regarding previous work is the use of target labels to choose the best model along training. This can be easily verified by inspecting the code. For PADA it can be seen on line 240 of the script "train_pada.py", for BA3US in line 116 for the script "run_partial.py", for M-POT it can be seen line 164 of the file "run_mOT.py", for SAFN it can be seen in the "eval.py" file and finally for AR in line 149 of the script "train.py".

For JUMBOT and M-POT which are based on optimal transport, we used the optimal transport solvers from (Flamary et al., 2021).

### A.2 Model Selection

DEV requires learning a discriminative model to distinguish source samples from target samples. Its neural network architecture must be specified as well the training details. You et al. (2019) (DEV) use a multilayer perceptron, while Saito et al. (2021) (SND) use a Support Vector Machine in their reimplementation of DEV. We empirically observed the latter to yield more stable weights and so that was the one we used. In order to train the SVM discriminator, following (Saito et al., 2021), we take 3000 feature embeddings from source samples used in training and 3000 random feature embeddings from target samples, both chosen randomly. We do a 80/20 split into training and test data. The SVM is trained with a linear kernel for a maximum of 4000 iterations. Of 5 different SVM models trained with decay values spaced evenly on log space between $10^{-2}$ and $10^4$ the one that leads to the highest accuracy (in distinguishing source from target features) on the test data split is the chosen one.

As for SND, it also requires specifying a temperature for temperature scaling component of the strategy. We used the default value of 0.05 that is suggested in (Saito et al., 2021).

Finally, we mention that the samples used for 100-RND were randomly selected and their list is made available together with the code. As for the samples used for 1-SHOT, they are the same as the ones used in semi-supervised domain adaptation.

### A.3  Optimizer

In general, all methods claim to adopt Nesterov's acceleration method as the optimization method with a momentum of 0.9 and setting the weight decay set to $5 \times 10^{-4}$. The learning rate follows the annealing strategy as in Ganin et al. (2016):

$$\mu_p = \mu_0(1 + \alpha \mathring{u} p)^{-\beta},$$

where $p$ is the training progress linearly changing from 0 to 1, $\mu_0 = 0.01$ and $\alpha = 10$ and $\beta = 0.75$.

However, inspecting the Official code repo for each PDA method, the actual learning schedule is given by

$$\mu_i = \mu_0(1 + \alpha \mathring{u} i)^{-\beta},$$

where $i$ is the iteration number in the training procedure, $\mu_0 = 0.01$ and $\alpha = 0.001$ and $\beta = 0.75$. Only when the total number of iterations is 10000 do the learning rate schedules match. In this work, we followed the latter since it is the one indeed used. For OFFICE-HOME, all methods are trained for 5000 iterations, while for VISDA they are trained for 10000 iterations, with the exception of the S. ONLY which is trained for 1000 iterations on OFFICE-HOME and 5000 iterations on VISDA.

### A.4  Hyper-Parameters

In Table 9, we report the values used for each hyper-parameter in our grid search. We report in Table 10 the hyper-parameters chosen by each model selection strategy for each method on both datasets. In addition, for the reproducibility of AR with the proposed architecture in Gu et al. (2021),a feature normalization layer is added in the bottleneck which requires specifying $r$, the value to which the 2-norm is set. This hyper-parameter is therefore included in the hyper-parameter grid search with the possible values of $\{5, 10, 20\}$ which are the different values used in the experiments in (Gu et al., 2021).

## B  Additional Discussion of Results

In this section, we provide additional results that we could not add to the main paper due to the space constraints.

We start by ensuring that our reimplementation of PDA methods was done correctly by comparing our reproduced results with previously reported results in Table 11 .

In Table 12, we show the accuracy per tasks on OFFICE-HOME averaged over three different seeds (2020, 2021, 2022) for all pairs of methods and model selection strategies.

In Table 13, we compare previously reported results with ours on VISDA. While proposed methods reported results on OFFICE-HOME, only PADA and AR results are reported in the original papers for VISDA. Gu et al. (2021) AR) also report results for BA3US. Analysing the results, we see a 9 percentage point decrease in average tasks accuracy for PADA, but our experiments show that there is a significant seed dependence which we discuss in detail below. This is particularly important since Cao et al. (2018) (PADA) report results from a single run. Comparing our best seeds for PADA on the SR and RS tasks, we achieve 58.01% and 67.9% accuracy versus a reported 53.53% and 76.5%. Moreover, we point out that the official code repository for PADA does not include the details to reproduce the VISDA experiments, so it is possible that minor tweaks (e.g learning rate) are necessary. As for BA3US, our results are within the standard deviation being better on the SR tasks and worse on the RS tasks. Finally as for AR we see a decrease in performance which, as the results on OFFICE-HOME show, can be explained by the differences in the neural network architecture.

Finally in Table 14, we show all the average tasks accuracies from all pairs of methods and model selection strategies on the OFFICE-HOME and VISDA datasets including the 50-RND model selection strategy.

| Method | HP | Values |
|---|---|---|
| PADA | $\lambda$ | $[0.1, 0.5, 1.0, 5.0, 10.0]$ |
| BA3US | $\lambda_{wce}$
$\lambda_{ent}$ | $[0.1, 0.5, 1, 5, 10]$
$[0.01, 0.05, 0.1, 0.5, 1]$ |
| SAFN | $\lambda$
$\Delta_r$ | $[0.005, 0.01, 0.05, 0.1, 0.5]$
$[0.01, 0.1, 1.0]$ |
| AR | $\rho_0$
$A_{up}$
$A_{low}$
$\lambda_{ent}$ | $[2.5, 5.0, 7.5, 10.0]$
$[5.0, 10.0]$
$-A_{up}$
$[0.01, 0.1, 1.0]$ |
| JUMBOT | $\tau$
$\eta_1$
$\eta_2$
$\eta_3$ | $[0.001, 0.01, 0.1]$
$[0.00001, 0.0001, 0.001, 0.01, 0.1]$
$[0.1, 0.5, 1.]$
$[5, 10, 20]$ |
| MPOT | $\epsilon$
$\eta_1$
$\eta_2$
$m$ | $[0.5, 1.0, 1.5]$
$[0.0001, 0.001, 0.01, 0.1, 1.0]$
$[0.1, 1.0, 5.0, 10.0]$
$[0.1, 0.2, 0.3, 0.4]$ |

Table 9: Hyper-Parameter values for each PDA method considered in the grid search.

| Method | Dataset | HP | ORACLE | 1-SHOT | 50-RND | 100-RND | S-ACC | ENT | DEV | SND |
|---|---|---|---|---|---|---|---|---|---|---|
| PADA | OFFICE-HOME | $\lambda$ | 0.5 | 0.1 | 0.1 | 0.5 | 0.1 | 1.0 | 5.0 | 0.5 |
| | VISDA | $\lambda$ | 0.5 | 1.0 | 10.0 | 0.5 | 1.0 | 0.5 | 5.0 | 0.1 |
| SAFN | OFFICE-HOME | $\lambda$ | 0.005 | 0.1 | 0.005 | 0.01 | 0.005 | 0.01 | 0.005 | 0.005 |
| | | $\Delta r$ | 0.1 | 0.01 | 0.01 | 0.01 | 0.01 | 0.1 | 0.1 | 0.1 |
| | VISDA | $\lambda$ | 0.005 | 0.005 | 0.05 | 0.05 | 0.005 | 0.05 | 0.005 | 0.05 |
| | | $\Delta r$ | 0.1 | 0.01 | 0.01 | 0.01 | 0.01 | 0.01 | 0.01 | 0.01 |
| BA3US | OFFICE-HOME | $\lambda_{wce}$ | 5.0 | 10.0 | 5.0 | 5.0 | 5.0 | 0.1 | 10.0 | 1.0 |
| | | $\lambda_{ent}$ | 0.05 | 0.05 | 0.01 | 0.05 | 0.01 | 0.1 | 0.05 | 0.01 |
| | VISDA | $\lambda_{wce}$ | 1.0 | 1.0 | 0.1 | 1.0 | 5.0 | 1.0 | 5.0 | 5.0 |
| | | $\lambda_{ent}$ | 0.5 | 0.5 | 0.5 | 0.5 | 0.05 | 0.5 | 0.05 | 1.0 |
| AR | OFFICE-HOME | $\rho_0$ | 2.5 | 2.5 | 5.0 | 5.0 | 2.5 | 5.0 | 7.5 | 10.0 |
| | | $A_{up}$ | 5.0 | 5.0 | 10.0 | 5.0 | 5.0 | 10.0 | 10.0 | 10.0 |
| | | $A_{low}$ | -5.0 | -5.0 | -10.0 | -5.0 | -5.0 | -10.0 | -10.0 | -10.0 |
| | | $\lambda_{ent}$ | 0.1 | 0.1 | 1.0 | 1.0 | 0.01 | 1.0 | 0.01 | 1.0 |
| | VISDA | $\rho_0$ | 2.5 | 2.5 | 2.5 | 2.5 | 2.5 | 7.5 | 2.5 | 10.0 |
| | | $A_{up}$ | 10.0 | 10.0 | 10.0 | 10.0 | 5.0 | 10.0 | 10.0 | 10.0 |
| | | $A_{low}$ | -10.0 | -10.0 | -10.0 | -10.0 | -5.0 | -10.0 | -10.0 | -10.0 |
| | | $\lambda_{ent}$ | 0.1 | 0.1 | 0.1 | 0.1 | 0.01 | 0.1 | 0.01 | 0.01 |
| JUMBOT | OFFICE-HOME | $\tau$ | 0.01 | 0.01 | 0.01 | 0.001 | 0.1 | 0.01 | 0.01 | 0.001 |
| | | $\eta_1$ | 0.0001 | 0.0001 | 0.001 | 0.0001 | 0.01 | 1e-05 | 0.01 | 1e-05 |
| | | $\eta_2$ | 0.5 | 1.0 | 0.5 | 0.1 | 0.1 | 0.5 | 1.0 | 1.0 |
| | | $\eta_3$ | 10.0 | 5.0 | 5.0 | 5.0 | 5.0 | 20.0 | 10.0 | 5.0 |
| | VISDA | $\tau$ | 0.01 | 0.01 | 0.01 | 0.01 | 0.001 | 0.01 | 0.001 | 0.01 |
| | | $\eta_1$ | 0.001 | 0.001 | 0.001 | 0.001 | 0.01 | 1e-05 | 0.01 | 0.0001 |
| | | $\eta_2$ | 1.0 | 1.0 | 0.5 | 1.0 | 0.1 | 0.5 | 1.0 | 1.0 |
| | | $\eta_3$ | 5.0 | 5.0 | 5.0 | 5.0 | 10.0 | 5.0 | 20.0 | 5.0 |
| MPOT | OFFICE-HOME | $\epsilon$ | 0.5 | 0.5 | 1.0 | 0.5 | 1.0 | 1.5 | 1.0 | 1.5 |
| | | $\eta_1$ | 0.01 | 0.01 | 0.01 | 0.01 | 0.001 | 0.0001 | 1.0 | 0.01 |
| | | $\eta_2$ | 10.0 | 1.0 | 1.0 | 1.0 | 1.0 | 10.0 | 0.1 | 1.0 |
| | | $m$ | 0.3 | 0.1 | 0.1 | 0.2 | 0.3 | 0.4 | 0.2 | 0.4 |
| | VISDA | $\epsilon$ | 0.5 | 0.5 | 0.5 | 0.5 | 1.0 | 1.0 | 1.0 | 0.5 |
| | | $\eta_1$ | 0.01 | 0.001 | 0.01 | 0.01 | 0.001 | 0.0001 | 0.0001 | 0.01 |
| | | $\eta_2$ | 1.0 | 1.0 | 1.0 | 1.0 | 1.0 | 10.0 | 1.0 | 10.0 |
| | | $m$ | 0.3 | 0.1 | 0.3 | 0.3 | 0.2 | 0.4 | 0.2 | 0.3 |

Table 10: Hyper-parameters selected for the different methods for each model selection strategy on both OFFICE-HOME and VISDA.

| Method | A2C | A2P | A2R | C2A | C2P | C2R | P2A | P2C | P2R | R2A | R2C | R2P | Avg |
|---|---|---|---|---|---|---|---|---|---|---|---|---|---|
| s. only[†] | 46.33 | 67.51 | 75.87 | 59.14 | 59.94 | 62.73 | 58.22 | 41.79 | 74.88 | 67.40 | 48.18 | 74.17 | 61.35 |
| s. only (Ours) | 45.43 | 68.91 | 79.53 | 55.59 | 57.42 | 65.23 | 59.32 | 40.80 | 75.80 | 69.88 | 47.20 | 77.31 | 61.87 |
| pada[†] | 51.95 | 67.00 | 78.74 | 52.16 | 53.78 | 59.03 | 52.61 | 43.22 | 78.79 | 73.73 | 56.60 | 77.09 | 62.06 |
| pada (Ours) | 50.53 | 67.45 | 80.14 | 57.30 | 54.47 | 64.55 | 61.07 | 40.94 | 79.55 | 73.09 | 54.63 | 80.93 | 63.72 |
| safn[†*] | 58.93 | 76.25 | 81.42 | 70.43 | 72.97 | 77.78 | 72.36 | 55.34 | 80.40 | 75.81 | 60.42 | 79.92 | 71.84 |
| safn* (Ours) | 59.98 | 79.85 | 85.18 | 72.02 | 73.73 | 78.54 | 76.09 | 59.32 | 83.25 | 80.04 | 64.20 | 84.44 | 74.72 |
| safn (Ours) | 49.57 | 68.55 | 78.26 | 57.91 | 59.29 | 66.81 | 59.87 | 45.29 | 75.98 | 69.08 | 51.68 | 77.29 | 63.30 |
| ba3us[†] | 60.62 | 83.16 | 88.39 | 71.75 | 72.79 | 83.40 | 75.45 | 61.59 | 86.53 | 79.25 | 62.80 | 86.05 | 75.98 |
| ba3us (Ours) | 63.26 | 82.75 | 89.16 | 69.91 | 71.93 | 77.58 | 75.73 | 59.94 | 86.89 | 80.93 | 66.77 | 86.93 | 75.98 |
| ar[†*] | 62.13 | 79.22 | 89.12 | 73.92 | 75.57 | 84.37 | 78.42 | 61.91 | 87.85 | 82.19 | 65.37 | 85.27 | 77.11 |
| ar* (Ours) | 62.75 | 81.55 | 89.07 | 71.63 | 73.41 | 82.94 | 75.88 | 61.03 | 85.70 | 79.86 | 62.93 | 85.30 | 76.00 |
| ar (Ours) | 57.33 | 79.61 | 86.31 | 69.45 | 71.88 | 79.94 | 70.28 | 53.57 | 83.78 | 77.26 | 59.68 | 83.72 | 72.73 |
| jumbot[†] | 62.70 | 77.50 | 84.40 | 76.00 | 73.30 | 80.50 | 74.70 | 60.80 | 85.10 | 80.20 | 66.50 | 83.90 | 75.47 |
| jumbot (Ours) | 61.87 | 78.19 | 88.11 | 77.69 | 76.75 | 84.15 | 76.83 | 63.72 | 84.80 | 81.79 | 64.70 | 87.17 | 77.15 |
| mpot[†] | 64.60 | 80.62 | 87.17 | 76.43 | 77.61 | 83.58 | 77.07 | 63.74 | 87.63 | 81.42 | 68.50 | 87.38 | 77.98 |
| mpot (Ours) | 64.48 | 80.88 | 86.78 | 76.22 | 77.95 | 82.59 | 75.18 | 64.60 | 84.87 | 80.59 | 67.04 | 86.52 | 77.31 |

Table 11: Comparison between reported (†) accuracies on partial OFFICE-HOME from published methods with our implementation using the ORACLE model selection strategy. * denotes different bottleneck architectures.

| TASK | METHOD | S-ACC | ENT | DEV | SND | 1-SHOT | 100-RND | ORACLE |
|---|---|---|---|---|---|---|---|---|
| AC | S. ONLY | 44.50 ± 1.7 | 45.27 ± 1.1 | 43.74 ± 1.8 | 42.23 ± 1.3 | 43.84 ± 1.7 | 43.28 ± 1.6 | 45.43 ± 0.9 |
| | PADA | 50.15 ± 2.8 | 46.03 ± 2.9 | 44.70 ± 1.3 | 50.43 ± 0.8 | 52.98 ± 0.2 | 50.41 ± 0.8 | 50.53 ± 0.7 |
| | SAFN | 47.36 ± 0.1 | 47.08 ± 2.0 | 48.12 ± 0.4 | 49.57 ± 0.3 | 31.40 ± 3.7 | 47.58 ± 0.8 | 49.57 ± 0.3 |
| | BA3US | 54.89 ± 4.7 | 59.26 ± 0.9 | 41.67 ± 18.9 | 62.21 ± 0.9 | 44.60 ± 21.0 | 62.53 ± 2.0 | 63.26 ± 1.0 |
| | AR | 51.12 ± 1.2 | 54.91 ± 1.8 | 49.25 ± 2.8 | 54.37 ± 1.6 | 56.00 ± 2.3 | 54.89 ± 2.0 | 57.33 ± 1.7 |
| | JUMBOT | 49.07 ± 0.2 | 57.69 ± 5.6 | 46.11 ± 0.1 | 56.60 ± 2.8 | 61.59 ± 1.7 | 61.07 ± 0.9 | 61.87 ± 1.4 |
| | MPOT | 53.07 ± 0.3 | 52.94 ± 2.0 | 46.07 ± 0.7 | 32.96 ± 0.4 | 53.97 ± 1.3 | 61.59 ± 1.2 | 64.48 ± 1.2 |
| AP | S. ONLY | 67.71 ± 2.4 | 68.91 ± 1.4 | 67.81 ± 1.2 | 68.91 ± 1.4 | 66.52 ± 3.1 | 68.76 ± 1.6 | 68.91 ± 1.4 |
| | PADA | 66.93 ± 1.2 | 62.09 ± 2.8 | 61.61 ± 5.4 | 66.72 ± 1.5 | 63.03 ± 1.6 | 67.21 ± 1.8 | 67.45 ± 1.6 |
| | SAFN | 66.82 ± 1.9 | 66.83 ± 0.5 | 67.30 ± 0.5 | 68.18 ± 1.3 | 49.73 ± 4.3 | 67.53 ± 0.8 | 68.55 ± 1.0 |
| | BA3US | 71.34 ± 0.8 | 76.38 ± 1.5 | 50.05 ± 28.7 | 83.29 ± 0.4 | 51.39 ± 29.8 | 82.09 ± 0.8 | 82.75 ± 0.9 |
| | AR | 72.79 ± 0.7 | 78.45 ± 1.8 | 70.20 ± 1.7 | 79.01 ± 2.2 | 78.58 ± 1.9 | 78.54 ± 1.4 | 79.61 ± 1.6 |
| | JUMBOT | 65.45 ± 0.4 | 75.44 ± 1.4 | 66.33 ± 0.6 | 68.48 ± 1.5 | 76.86 ± 3.4 | 77.87 ± 1.4 | 78.19 ± 2.4 |
| | MPOT | 72.61 ± 1.2 | 68.94 ± 1.2 | 65.43 ± 0.8 | 49.73 ± 1.1 | 68.78 ± 1.7 | 75.56 ± 1.7 | 80.88 ± 3.3 |
| AR | S. ONLY | 78.37 ± 0.3 | 79.26 ± 0.7 | 78.28 ± 0.7 | 79.35 ± 0.6 | 77.38 ± 0.9 | 77.97 ± 1.2 | 79.53 ± 0.3 |
| | PADA | 76.73 ± 1.7 | 76.05 ± 1.4 | 68.99 ± 11.3 | 79.72 ± 1.8 | 78.06 ± 2.6 | 79.97 ± 1.5 | 80.14 ± 1.4 |
| | SAFN | 77.62 ± 0.2 | 77.73 ± 0.2 | 77.43 ± 0.5 | 77.86 ± 0.5 | 62.82 ± 2.0 | 77.91 ± 0.4 | 78.26 ± 0.2 |
| | BA3US | 81.91 ± 3.9 | 86.03 ± 0.6 | 63.74 ± 26.1 | 88.50 ± 0.6 | 65.47 ± 27.2 | 88.28 ± 0.4 | 89.16 ± 0.2 |
| | AR | 77.91 ± 0.2 | 84.23 ± 0.9 | 79.73 ± 2.5 | 84.54 ± 0.8 | 82.77 ± 2.0 | 84.34 ± 0.6 | 86.31 ± 0.4 |
| | JUMBOT | 77.14 ± 0.3 | 85.24 ± 2.7 | 76.42 ± 0.3 | 84.70 ± 2.1 | 86.45 ± 2.1 | 86.01 ± 1.3 | 88.11 ± 1.5 |
| | MPOT | 78.50 ± 0.7 | 75.98 ± 0.6 | 76.46 ± 0.4 | 57.39 ± 1.4 | 78.04 ± 2.1 | 82.59 ± 0.6 | 86.78 ± 0.5 |
| CA | S. ONLY | 52.56 ± 0.9 | 54.21 ± 2.1 | 51.42 ± 2.7 | 51.76 ± 3.7 | 50.47 ± 2.4 | 53.75 ± 1.1 | 55.59 ± 0.7 |
| | PADA | 58.00 ± 1.4 | 55.07 ± 2.7 | 35.08 ± 13.1 | 57.30 ± 1.9 | 51.67 ± 5.0 | 56.69 ± 1.5 | 57.30 ± 1.9 |
| | SAFN | 57.85 ± 0.6 | 56.54 ± 2.2 | 56.75 ± 0.3 | 57.91 ± 0.3 | 48.88 ± 2.4 | 56.47 ± 1.0 | 57.91 ± 0.3 |
| | BA3US | 61.68 ± 5.2 | 68.96 ± 1.8 | 60.70 ± 2.2 | 68.50 ± 0.9 | 65.63 ± 1.4 | 69.15 ± 1.2 | 69.91 ± 0.2 |
| | AR | 63.21 ± 1.5 | 64.86 ± 2.3 | 62.72 ± 1.0 | 64.52 ± 1.6 | 68.99 ± 0.2 | 64.95 ± 2.4 | 69.45 ± 0.5 |
| | JUMBOT | 60.09 ± 0.1 | 75.97 ± 1.4 | 56.81 ± 0.1 | 71.81 ± 1.8 | 74.20 ± 0.9 | 74.56 ± 0.4 | 77.69 ± 0.1 |
| | MPOT | 61.92 ± 0.5 | 60.58 ± 0.8 | 56.44 ± 1.0 | 44.11 ± 2.4 | 69.24 ± 0.4 | 72.48 ± 1.0 | 76.22 ± 0.1 |
| CP | S. ONLY | 54.81 ± 0.1 | 55.52 ± 0.6 | 54.55 ± 1.2 | 53.48 ± 2.1 | 53.24 ± 2.0 | 55.57 ± 2.2 | 57.42 ± 1.2 |
| | PADA | 56.13 ± 1.4 | 47.28 ± 0.1 | 24.24 ± 20.5 | 52.10 ± 1.7 | 56.28 ± 0.4 | 53.86 ± 1.6 | 54.47 ± 1.7 |
| | SAFN | 57.89 ± 0.7 | 59.07 ± 0.7 | 58.17 ± 1.2 | 58.17 ± 1.2 | 45.27 ± 0.7 | 58.19 ± 0.4 | 59.29 ± 0.5 |
| | BA3US | 67.13 ± 3.9 | 71.07 ± 0.8 | 59.08 ± 10.9 | 71.45 ± 3.6 | 59.78 ± 15.3 | 71.65 ± 1.5 | 71.93 ± 1.6 |
| | AR | 60.54 ± 4.0 | 68.16 ± 3.5 | 61.85 ± 4.6 | 68.05 ± 3.2 | 68.35 ± 1.9 | 69.00 ± 3.7 | 71.88 ± 0.9 |
| | JUMBOT | 59.59 ± 1.3 | 74.85 ± 3.3 | 56.36 ± 0.5 | 71.84 ± 1.6 | 73.43 ± 3.3 | 76.40 ± 1.4 | 76.75 ± 0.8 |
| | MPOT | 64.16 ± 1.8 | 65.99 ± 2.2 | 57.95 ± 1.0 | 38.66 ± 1.2 | 65.88 ± 0.5 | 69.77 ± 0.9 | 77.95 ± 1.3 |
| CR | S. ONLY | 62.88 ± 0.9 | 63.19 ± 0.3 | 63.94 ± 1.7 | 63.94 ± 1.7 | 61.77 ± 1.1 | 63.94 ± 0.4 | 65.23 ± 0.8 |
| | PADA | 66.45 ± 0.8 | 60.92 ± 2.4 | 61.66 ± 2.4 | 63.11 ± 1.9 | 64.00 ± 1.4 | 63.94 ± 1.3 | 64.55 ± 1.1 |
| | SAFN | 66.92 ± 0.9 | 66.22 ± 0.5 | 65.64 ± 1.3 | 66.13 ± 1.0 | 57.26 ± 2.2 | 65.88 ± 0.2 | 66.81 ± 0.5 |
| | BA3US | 72.96 ± 1.0 | 76.22 ± 1.2 | 67.88 ± 0.9 | 76.96 ± 0.6 | 68.49 ± 1.3 | 77.21 ± 0.6 | 77.58 ± 0.9 |
| | AR | 72.76 ± 0.9 | 80.45 ± 0.8 | 70.86 ± 5.6 | 79.16 ± 2.8 | 77.25 ± 1.4 | 79.57 ± 0.2 | 79.94 ± 0.8 |
| | JUMBOT | 66.67 ± 1.3 | 79.75 ± 1.2 | 66.70 ± 0.8 | 80.91 ± 0.9 | 79.85 ± 0.3 | 81.54 ± 1.7 | 84.15 ± 1.3 |
| | MPOT | 70.22 ± 0.2 | 71.51 ± 0.8 | 66.35 ± 1.0 | 50.06 ± 1.0 | 71.42 ± 0.7 | 75.41 ± 0.5 | 82.59 ± 0.7 |
| PA | S. ONLY | 58.77 ± 0.5 | 56.96 ± 1.5 | 57.94 ± 0.9 | 55.37 ± 0.6 | 56.11 ± 1.7 | 58.37 ± 0.4 | 59.32 ± 0.7 |
| | PADA | 60.33 ± 2.1 | 56.69 ± 2.8 | 57.91 ± 1.7 | 60.82 ± 3.0 | 58.92 ± 3.3 | 60.27 ± 2.7 | 61.07 ± 3.0 |
| | SAFN | 58.80 ± 0.7 | 56.75 ± 2.1 | 59.08 ± 0.5 | 59.14 ± 0.8 | 42.33 ± 1.6 | 59.69 ± 0.1 | 59.87 ± 0.7 |
| | BA3US | 68.90 ± 5.0 | 73.16 ± 0.6 | 64.62 ± 1.6 | 76.19 ± 1.2 | 68.38 ± 1.7 | 75.15 ± 1.3 | 75.73 ± 1.3 |
| | AR | 63.39 ± 3.1 | 67.58 ± 0.4 | 61.65 ± 1.0 | 65.60 ± 1.7 | 69.67 ± 1.5 | 66.73 ± 0.3 | 70.28 ± 1.0 |
| | JUMBOT | 60.24 ± 1.0 | 72.85 ± 2.4 | 58.03 ± 1.1 | 70.28 ± 0.8 | 74.96 ± 3.4 | 72.60 ± 1.2 | 76.83 ± 1.9 |
| | MPOT | 64.13 ± 0.9 | 58.28 ± 0.9 | 57.64 ± 0.8 | 43.74 ± 4.3 | 70.31 ± 1.0 | 72.64 ± 0.9 | 75.18 ± 1.3 |
| PC | S. ONLY | 39.28 ± 0.8 | 38.75 ± 0.6 | 39.40 ± 0.9 | 37.35 ± 1.0 | 37.35 ± 1.0 | 39.12 ± 0.4 | 40.80 ± 0.9 |
| | PADA | 43.50 ± 1.2 | 38.43 ± 3.0 | 38.03 ± 0.6 | 39.26 ± 2.0 | 43.62 ± 1.0 | 40.56 ± 1.8 | 40.94 ± 1.6 |
| | SAFN | 42.49 ± 0.6 | 39.58 ± 2.0 | 43.00 ± 1.1 | 43.90 ± 0.5 | 29.77 ± 2.6 | 43.14 ± 1.7 | 45.29 ± 0.7 |
| | BA3US | 55.92 ± 1.3 | 57.91 ± 2.5 | 56.74 ± 1.3 | 59.94 ± 1.7 | 57.83 ± 1.3 | 58.17 ± 1.0 | 59.94 ± 0.7 |
| | AR | 48.36 ± 1.7 | 52.34 ± 1.0 | 43.72 ± 0.7 | 51.28 ± 1.6 | 51.98 ± 1.8 | 50.85 ± 1.4 | 53.57 ± 0.2 |
| | JUMBOT | 43.60 ± 0.0 | 60.18 ± 0.9 | 41.99 ± 0.8 | 50.69 ± 4.9 | 62.87 ± 0.6 | 59.92 ± 0.4 | 63.72 ± 0.5 |
| | MPOT | 50.87 ± 1.1 | 49.87 ± 2.6 | 43.60 ± 0.6 | 28.66 ± 2.6 | 53.03 ± 0.7 | 57.67 ± 1.6 | 64.60 ± 0.0 |
| PR | S. ONLY | 75.08 ± 0.5 | 75.65 ± 1.3 | 74.91 ± 0.6 | 74.10 ± 2.8 | 71.97 ± 1.8 | 75.56 ± 1.3 | 75.80 ± 1.2 |
| | PADA | 76.70 ± 0.4 | 77.08 ± 0.2 | 73.11 ± 3.4 | 79.33 ± 1.3 | 74.27 ± 4.1 | 78.91 ± 1.8 | 79.55 ± 1.4 |
| | SAFN | 75.46 ± 0.4 | 73.90 ± 0.9 | 74.64 ± 0.4 | 75.81 ± 0.7 | 63.52 ± 3.2 | 75.00 ± 0.7 | 75.98 ± 0.6 |
| | BA3US | 79.13 ± 4.7 | 85.59 ± 1.2 | 75.21 ± 0.6 | 86.31 ± 1.4 | 82.05 ± 1.0 | 85.92 ± 1.3 | 86.89 ± 0.5 |
| | AR | 78.02 ± 1.7 | 82.48 ± 1.9 | 76.29 ± 0.7 | 83.05 ± 1.1 | 78.72 ± 1.0 | 82.39 ± 1.9 | 83.78 ± 1.0 |
| | JUMBOT | 74.43 ± 0.9 | 83.21 ± 1.1 | 74.97 ± 0.5 | 83.89 ± 1.5 | 81.83 ± 0.9 | 84.63 ± 2.3 | 84.80 ± 1.3 |
| | MPOT | 77.40 ± 0.1 | 73.77 ± 1.3 | 74.86 ± 1.3 | 58.40 ± 1.9 | 76.88 ± 1.3 | 82.02 ± 0.6 | 84.87 ± 1.4 |
| RA | S. ONLY | 68.90 ± 0.6 | 69.24 ± 1.0 | 69.27 ± 1.0 | 68.53 ± 1.4 | 68.96 ± 0.5 | 69.02 ± 0.5 | 69.88 ± 0.9 |
| | PADA | 69.27 ± 3.5 | 69.48 ± 1.3 | 66.33 ± 0.7 | 73.09 ± 1.5 | 68.26 ± 3.1 | 72.70 ± 1.4 | 73.09 ± 1.5 |
| | SAFN | 67.92 ± 0.0 | 67.80 ± 0.2 | 68.11 ± 0.9 | 68.17 ± 1.6 | 56.11 ± 3.2 | 69.64 ± 1.0 | 69.08 ± 0.6 |
| | BA3US | 72.27 ± 3.5 | 78.11 ± 1.4 | 70.92 ± 2.0 | 79.46 ± 1.4 | 80.78 ± 1.1 | 79.86 ± 2.1 | 80.93 ± 0.8 |
| | AR | 70.00 ± 1.1 | 74.75 ± 2.1 | 70.31 ± 1.7 | 75.02 ± 1.6 | 76.19 ± 0.7 | 74.66 ± 2.3 | 77.26 ± 0.6 |
| | JUMBOT | 70.19 ± 0.5 | 81.97 ± 1.0 | 67.43 ± 0.3 | 81.21 ± 0.6 | 78.48 ± 2.0 | 81.85 ± 1.7 | 81.79 ± 0.8 |
| | MPOT | 70.40 ± 0.6 | 64.98 ± 0.4 | 67.68 ± 0.5 | 56.90 ± 1.9 | 76.52 ± 0.4 | 79.80 ± 0.5 | 80.59 ± 0.6 |
| RC | S. ONLY | 45.33 ± 1.0 | 45.31 ± 1.0 | 45.33 ± 1.0 | 43.78 ± 0.6 | 46.13 ± 2.0 | 43.46 ± 0.2 | 47.20 ± 0.9 |
| | PADA | 53.93 ± 1.3 | 49.73 ± 3.5 | 29.97 ± 21.0 | 45.77 ± 1.6 | 54.25 ± 1.6 | 53.39 ± 2.2 | 54.63 ± 0.9 |
| | SAFN | 49.73 ± 0.1 | 48.76 ± 0.1 | 50.53 ± 0.6 | 49.59 ± 1.6 | 37.55 ± 0.8 | 50.85 ± 0.3 | 51.68 ± 0.8 |
| | BA3US | 51.84 ± 0.5 | 62.85 ± 2.7 | 58.39 ± 2.3 | 65.35 ± 1.9 | 63.10 ± 0.8 | 66.57 ± 1.5 | 66.77 ± 1.5 |
| | AR | 52.52 ± 1.0 | 55.64 ± 1.2 | 49.61 ± 0.8 | 55.02 ± 1.8 | 55.48 ± 2.1 | 55.42 ± 1.6 | 59.68 ± 1.1 |
| | JUMBOT | 51.12 ± 1.1 | 61.81 ± 4.6 | 48.12 ± 0.5 | 58.85 ± 1.7 | 61.59 ± 2.2 | 64.84 ± 1.0 | 64.70 ± 1.1 |
| | MPOT | 53.99 ± 1.5 | 57.53 ± 0.6 | 48.12 ± 0.8 | 39.34 ± 1.2 | 57.39 ± 1.7 | 64.64 ± 0.1 | 67.04 ± 0.6 |
| RP | S. ONLY | 76.34 ± 0.7 | 76.47 ± 0.8 | 75.99 ± 1.3 | 75.84 ± 1.6 | 73.33 ± 2.2 | 75.28 ± 2.3 | 77.31 ± 0.1 |
| | PADA | 78.88 ± 0.8 | 78.00 ± 1.7 | 71.07 ± 11.3 | 80.62 ± 0.4 | 78.62 ± 0.4 | 80.73 ± 0.9 | 80.93 ± 0.6 |
| | SAFN | 76.23 ± 0.8 | 76.23 ± 0.7 | 75.65 ± 0.5 | 76.64 ± 0.5 | 67.00 ± 1.4 | 76.41 ± 0.8 | 77.29 ± 0.5 |
| | BA3US | 81.85 ± 4.1 | 84.84 ± 0.6 | 78.06 ± 1.3 | 86.35 ± 0.9 | 79.20 ± 1.1 | 85.66 ± 1.0 | 86.93 ± 0.2 |
| | AR | 77.55 ± 2.6 | 83.06 ± 1.2 | 75.61 ± 0.4 | 83.40 ± 0.9 | 82.73 ± 1.0 | 82.80 ± 0.4 | 83.72 ± 0.6 |
| | JUMBOT | 77.12 ± 1.3 | 86.33 ± 1.6 | 76.04 ± 0.1 | 88.18 ± 0.4 | 87.34 ± 0.2 | 87.64 ± 0.7 | 87.17 ± 1.7 |
| | MPOT | 77.61 ± 0.3 | 73.17 ± 2.7 | 75.89 ± 0.4 | 63.14 ± 0.6 | 77.95 ± 1.4 | 82.60 ± 0.5 | 86.52 ± 1.2 |
| Avg | S. ONLY | 60.38 ± 0.5 | 60.73 ± 0.2 | 60.22 ± 0.3 | 59.55 ± 0.3 | 58.92 ± 0.4 | 60.34 ± 0.4 | 61.87 ± 0.3 |
| | PADA | 63.08 ± 0.3 | 59.74 ± 0.5 | 52.72 ± 2.8 | 62.36 ± 0.4 | 62.00 ± 0.5 | 63.22 ± 0.1 | 63.72 ± 0.3 |
| | SAFN | 62.09 ± 0.2 | 61.37 ± 0.3 | 62.03 ± 0.4 | 62.59 ± 0.1 | 49.30 ± 0.7 | 62.36 ± 0.2 | 63.30 ± 0.2 |
| | BA3US | 68.32 ± 1.1 | 73.36 ± 0.6 | 62.25 ± 7.1 | 75.37 ± 0.8 | 65.56 ± 7.6 | 75.19 ± 0.4 | 75.98 ± 0.3 |
| | AR | 65.68 ± 0.3 | 70.58 ± 0.4 | 64.32 ± 0.9 | 70.25 ± 0.2 | 70.56 ± 0.7 | 70.34 ± 0.2 | 72.73 ± 0.3 |
| | JUMBOT | 62.89 ± 0.2 | 74.61 ± 0.8 | 61.28 ± 0.1 | 72.29 ± 0.2 | 74.95 ± 0.1 | 75.74 ± 0.3 | 77.15 ± 0.4 |
| | MPOT | 66.24 ± 0.1 | 64.46 ± 0.1 | 61.37 ± 0.2 | 46.92 ± 0.4 | 68.28 ± 0.2 | 73.06 ± 0.3 | 77.31 ± 0.5 |

Table 12: Average accuracy of different PDA methods based on different model selection strategies on the 12 tasks of Partial OFFICE-HOME. Average is done over three seeds (2020, 2021, 2022). For each method, we highlight the best and worst label-free model selection strategies in green and red, respectively.

| ALGORITHM | S2R | R2S | Avg |
|---|---|---|---|
| S. ONLY[†] | 45.26 | 64.28 | 54.77 |
| S. ONLY (Ours) | 51.86 | 67.11 | 59.48 |
| PADA[†] | 53.53 | 76.50 | 65.02 |
| PADA (Ours) | 49.34 | 59.81 | 54.57 |
| SAFN[†] | 67.65 | - | - |
| SAFN (Ours) | 56.88 | 68.40 | 62.64 |
| BA3US[†] | 69.86 | 67.56 | 68.71 |
| BA3US (Ours) | 71.77 | 63.56 | 67.67 |
| AR[†*] | 85.30 | 74.82 | 80.06 |
| AR (Ours) | 76.33 | 71.36 | 73.85 |
| JUMBOT[†] | - | - | - |
| JUMBOT (Ours) | 90.55 | 77.46 | 84.01 |
| MPOT[†] | - | - | - |
| MPOT (Ours) | 87.23 | 86.67 | 86.95 |

Table 13: Comparison between reported (†) accuracies on partial VISDA from published methods with our implementation using the ORACLE model selection strategy. * denotes different bottleneck architectures.

| DATASET | METHOD | S-ACC | ENT | DEV | SND | 1-SHOT | 50-RND | 100-RND | ORACLE |
|---|---|---|---|---|---|---|---|---|---|
| OFFICE-HOME | S. ONLY | 60.38±0.5 | 60.73±0.2 | 60.22±0.3 | 59.55±0.3 | 58.92±0.4 | 60.28±0.4 | 60.34±0.4 | 61.87±0.3 |
| | PADA | 63.08±0.3 | 59.74±0.5 | 52.72±2.8 | 62.36±0.4 | 62.00±0.5 | 63.82±0.4 | 63.22±0.1 | 63.72±0.3 |
| | SAFN | 62.09±0.2 | 61.37±0.3 | 62.03±0.4 | 62.59±0.1 | 49.30±0.7 | 62.00±0.2 | 62.36±0.2 | 63.30±0.2 |
| | BA3US | 68.32±1.1 | 73.36±0.6 | 62.25±7.1 | 75.37±0.8 | 65.56±7.6 | 73.22±0.3 | 75.19±0.4 | 75.98±0.3 |
| | AR | 65.68±0.3 | 70.58±0.4 | 64.32±0.9 | 70.25±0.2 | 70.56±0.7 | 70.26±0.2 | 70.34±0.2 | 72.73±0.3 |
| | JUMBOT | 62.89±0.2 | 74.61±0.8 | 61.28±0.1 | 72.29±0.2 | 74.95±0.1 | 64.95±0.3 | 75.74±0.3 | 77.15±0.4 |
| | MPOT | 66.24±0.1 | 64.46±0.1 | 61.37±0.2 | 46.92±0.4 | 68.28±0.2 | 69.90±0.5 | 73.06±0.3 | 77.31±0.5 |
| VISDA | S. ONLY | 55.15±2.4 | 55.24±3.2 | 55.07±1.2 | 55.02±2.9 | 55.72±2.2 | 57.90±1.1 | 58.16±0.6 | 59.48±0.4 |
| | PADA | 47.48±4.8 | 32.32±4.9 | 43.43±5.3 | 56.83±1.0 | 53.15±2.9 | 55.67±2.5 | 54.38±2.7 | 54.57±2.6 |
| | SAFN | 58.20±1.7 | 42.83±6.3 | 58.62±1.3 | 44.82±8.8 | 56.89±2.1 | 57.90±3.3 | 59.09±2.8 | 62.64±1.5 |
| | BA3US | 55.10±3.7 | 65.58±1.4 | 58.40±1.4 | 51.07±4.3 | 64.77±1.4 | 66.66±2.4 | 67.44±1.2 | 67.67±1.3 |
| | AR | 66.68±1.0 | 64.27±3.6 | 67.20±1.5 | 55.69±0.9 | 70.29±1.7 | 71.91±0.3 | 72.60±0.8 | 73.85±0.9 |
| | JUMBOT | 60.63±0.7 | 62.42±2.4 | 59.86±0.6 | 77.69±4.2 | 78.34±1.9 | 82.85±2.9 | 83.49±1.9 | 84.01±1.9 |
| | MPOT | 70.02±2.0 | 74.64±4.4 | 61.62±1.3 | 78.40±3.9 | 70.96±3.7 | 86.65±5.1 | 86.69±5.1 | 86.95±5.0 |

Table 14: Task accuracy average over seeds 2020, 2021, 2022 on Partial OFFICE-HOME and Partial VISDA for the PDA methods and model selection strategies. For each method, we highlight the best and worst label-free model selection strategies in green and red, respectively.

