# OpenReview forum: "A Reproducible and Realistic Evaluation of Partial Domain Adaptation Methods"
_TMLR — Accepted by TMLR_

### Review · Reviewer_fsHm · 2023-02-26

**Summary Of Contributions:**

This paper performs a thorough experimental evaluation of methods for Partial Domain Adaptation (PDA) (a variant of Domain Adaptation (DA) where the target domain contains a strict subset of the classes appearing in the source domain, making the problem harder compared to standard DA). In particular, their evaluation is realistic in the sense that they relax the optimistic assumption made in previous work that a labeled set of examples are available from the target dataset for model selection / validation (as this assumption violates the definition of DA’s problem setup, that assumes only unlabeled examples from the target domain). They consider cases where a small(er) number of labeled examples are available from the target domain for validation (which might be viable in practice), as well as several more ‘legal’ variants for model selection without a labeled validation set, that were proposed in some previous work. Their thorough analysis surfaces interesting findings about the generalizability of model selection approaches, the degree of optimism that results from using an ‘oracle’ for model selection (that uses labeled target examples), the reliance of results on the random seed, and other findings in the context of specific PDA methods evaluated and their reliance on hyperparameters like architectural choices. Finally, the authors developed and released a framework in pytorch, BenchmarkPDA, for facilitating research in this direction.

**Audience:**

Yes

**Broader Impact Concerns:**

I have no broader impact concerns.

**Claims And Evidence:**

Yes

**Requested Changes:**

- Add a formal problem description of the problem setting of PDA
- Add motivation for why the scope is limited to PDA in this work: why PDA in particular instead of other variants of the problem, and why not DA more generally
- Add motivations for why the particular PDA methods were chosen for evaluation.
- It would be great to also add more detail about the PDA methods and the model selection methods evaluated coming from previous literature. While the high-level descriptions provided are nice, more details into each approach would be useful to help the reader engage better with the space of solutions which in turn would aid in understanding and interpreting the findings from this evaluation
- Add discussion on whether the findings from this work are analogous to or contradict findings in previous studies of similar spirit (e.g. Gulrajani and Paz for domain generalization). It sounds like that study and problem setting is quite related. Though I understand domain generalization does not even assume unlabelled examples from the target domain, some model selection approaches can be applied to both, like using a subset of the source for validation and the oracle. How do results differ w.r.t these, for example, across the two studies?
- Define ‘negative transfer’, as the term appears several times in the paper, e.g. in the context of explaining why PDA is harder than DA. I’ve seen this term used before in transfer learning / continual learning to describe the issue arising when solving an earlier problem makes a later problem harder to solve. But here, there is no sequential task solving, if I understand correctly: there is just 1 training phase where the model is trained on labeled source data and unlabeled target data simultaneously (though, it would be nice to have a formal problem description as mentioned above, to clarify this point).
- In Section 4.2, the authors mention ‘... for each task of each dataset’. The term ‘task’ hadn’t been used up to this point (unless I missed it?) and is not clear what it means here (I can somewhat infer from context, but should be defined)
- In the same section,  the authors say that the unsuccessful runs of the hyperparameter search pose a challenge for DEV, SND and ENT, but it is not clear why that is. Looking at the description of these methods in Section 2, it seems that there are some missing pieces in terms of understanding why this is an issue. Please elaborate.
- Regarding hyperparameters, the authors say ‘we only consider the model at the end of training’. This confused me for two reasons: first, isn’t the ‘end of training’ also a hyperparameter in and of itself (i.e the number of steps)? If not, then what is the stopping criterion? Second, I’m a bit confused here because, earlier (Section 4.1, in the ‘Model Selection Strategies’ paragraph), the authors said ‘we use them [model selection approaches] both for hyper-parameter tuning as well selecting the best model along training’. This seems to contradict the statement of only considering the model at the end of training? It would be great to elaborate on what exactly is the procedure for choosing hyperparameters and a checkpoint
- In Section 4.2, the authors also say: ‘Then, to obtain the results with our neural network architecture on all tasks of each dataset, we trained an additional …’. It wasn’t clear to me what this additional training phase is for. I thought that after model selection, the chosen model (i.e. set of hyperparameters and checkpoint) would be directly deployed on the test set? The ‘with our neural network architecture’ part makes it sound like a different architecture was used for tuning rather than testing? But that doesn’t sound right, so I think I’m misunderstanding the setup. Please explain.
- Section 5 about reproducibility: by reading the discussion here I realized that the reproduced numbers are sometimes obtained with a different architecture compared to the original numbers compared against? I wouldn’t really call this a reproducibility trial then, as the architecture is an important component which is changed. For the sake of verifying reproducibility / correctness of implementation, it would make more sense to use the same setup (including the same architecture) as the one that was used to obtain the results we are trying to reproduce.
- On a similar vein, it sounds like the authors have changed the architectures of some of the compared approaches, in order to have the same architecture used for all. While I do agree that it’s nice to eliminate that factor of variation, in order to directly compare methods to each other, it might be that the performance of some is underestimated due to it being strongly dependent on a particular architecture from which it has now been decoupled? To what extent do the authors think it’s a possibility? Perhaps for the purposes of this paper this is OK but it would be great to add some discussion about the pros and cons of this design choice.
- RE: PADA combined with SND ‘performs better than the oracle on average, although within the standard deviation’: my understanding is that for assessing statistical significance, one needs to look at overlap of confidence intervals, not standard deviations. I find the wording here confusing.
- Table 7: state what is the relationship between this and Table 6 (e.g. in the caption). Currently the captions of these two tables are similar and it takes some scrolling up and down to find the differences.
- Section 5.3, when discussing random seeds, please state what exactly these seeds control (e.g. is it just the order of mini batches, or something more? Is this method-specific?)
- Minor: fix inconsistent notation, e.g. dev (in section 2) versus DEV elsewhere in the paper; analogously for ent, snd, etc

Additional comments and discussion
- Table 4 is hard to read as it contains a large number of figures. Maybe a better presentation of these results would be through a (set of) barplots, where each one shows the difference between the published results and reproduced results?
- Regarding experimental setup, as explained in Section 4.1 for the datasets, can the particular selection of the first X categories to serve as the partial target domain bias the results? That is, conceivably the PDA problem becomes easier or harder depending on which categories are chosen to be source-only (excluded from the target). Is there a rationale for choosing the first X? Has previous literature studied this?


**Strengths And Weaknesses:**

Strengths
- The paper studies an important topic which often receives too little attention in papers presenting new methods for problem settings that involve generalization outside of the training distribution without labeled examples. As the authors point out, the unavailability of labeled examples from the target distribution also implies the unavailability of a validation set for model selection, therefore model selection becomes an open problem in this space too.
- Thorough empirical investigation of a large number of methods and model selection approaches on two different datasets, with design choices made to enable fair comparisons and reproducible results
- Interesting findings and recommendations for the (P)DA community
- The paper is nicely written and easy to follow (see below for some small exceptions)

Weaknesses
- Unclear (missing motivation) why the scope is limited to PDA in particular instead of DA more generally or other variants of DA
- Missing formal problem description of DPA
- Insufficient description of previous methods (both in DA/PDA methods as well as previous approaches to model selection considered)
- Insufficient discussion of some design choices, as well as on the relationship between the findings of this work and findings of related papers (see below)
- Some places where clarity can be improved (see below)

---

> ### Author Response · Authors · 2023-03-13
> **Thank you for your constructive criticism. (1/3)**
>
> We thank the reviewer for acknowledging the interest of our study on the model selection strategies in Partial Domain Adaptation. We also thank the reviewer for the thorough review. Below we answer each of the raised concerns. We have updated our manuscript and all changes can be seen in blue.
>
>
> - Reviewer fsHm: "Unclear (missing motivation) why the scope is limited to PDA in particular instead of DA more generally or other variants of DA"
>
> The reviewer makes an interesting point. It is true that several other domain adaptation variants could have been considered. However, we would like to point out that the standard domain adaptation setting has been studied in concurrent work [1]. Furthermore, label shift between domains is typical in real-world data. The partial domain adaptation setting corresponds to an extreme label shift scenario where some source classes are not in the target domain, and it needs dedicated solutions as typical domain adaptation methods do not perform well on the PDA setting. More generally, other DA variants also require dedicated solutions to handle the considered DA scenario. We believe that it would have been hard to test several methods with several model selection strategies in the same paper for all domain adaptation variants. That is why we preferred tackling extensively only one domain adaptation setting. We have added a paragraph in the introduction explaining our choice.
>
> On a final note, addressing the other variants would have required significantly more experiments which we are unable to tackle with the computational resources available to us (8 NVIDIA GeForce RTX 2080 Ti). Thus, we decided to focus on the PDA variant given our expertise in this domain and given that previous work that proposes model selection strategies ([2, 3]) focus its attention on standard domain adaptation claiming their strategies to work on PDA but doing so based on limited experiments.
>
>
> [1] Musgrave et al, Three New Validators and a Large-Scale Benchmark Ranking for Unsupervised Domain Adaptation
>
> [2] You et al, Towards Accurate Model Selection in Deep Unsupervised Domain Adaptation
>
> [3] Saito et al, Tune it the Right Way: Unsupervised Validation of Domain Adaptation via Soft Neighborhood Density
>
> ---------------------
>
> - Reviewer fsHm: "Missing formal problem description of DPA"
>
> Thank you for your suggestion. We added the formal description of DA and PDA settings in Section 2.
>
> ---------------------
>
> - Reviewer fsHm: "Insufficient description of previous methods"
>
> We have added formal descriptions of the adversarial training formulation and divergence minimization that we consider in Section 3.
>
> ---------------------
>
> - Reviewer fsHm: "Add motivations for why the particular PDA methods were chosen for evaluation."
>
> Methods to address partial domain adaptation can be grouped into two categories: adversarial training or divergence minimization. We considered 3 methods from each approach. PADA, BA3US and AR rely on the adversarial training. PADA and BA3US are based on the well known DANN method, while AR was designed based on WGAN. Regarding the divergence minomization methods, we considered SAFN, JUMBOT and MPOT. The two last methods rely on the minimization of optimal transport costs, while SAFN minimizes the Maximum Mean Discrepancy.
>
> ---------------------
>
> - Reviewer fsHm: "It would be great to also add more detail about the PDA methods (...) understanding and interpreting the findings from this evaluation"
>
> Thank you for your suggestion. We have added a formal description of each model selection strategy and we have added the formal setting for each PDA method.
>
> ---------------------
>
> - Reviewer fsHm: "Add discussion on whether the findings from this work are analogous to or contradict findings in previous studies (...)"
>
> Gulrajani & Lopez-Paz (2021) for domain generalization and Saito et al. (2021) for unsupervised domain adaptation argue that the methods remain incomplete without model selection strategies as the latter can have a big impact on their performance. Our work is in agreement with this conclusion. We briefly mentioned this in the introduction, but have now added a longer discussion in Section 5.
>
> ---------------------
>
> - Reviewer fsHm: "Define ‘negative transfer'(...)".
>
> Thank you for pointing out the lack of definitions. We initially thought it was a common term in the DA literature but it seems that is not entirely true. In order to ease the burden of the reader, we have replaced this term with what it means, transferring source-only samples to the target domain.
>
> ---------------------
>
> - Reviewer fsHm: "The term ‘task’ hadn’t been used up to this point (...)"
>
> Thank you for your comment. A task refers to a domain adaptation scenario where one domain of the dataset is chosen as source and another one is chosen as target. We added this definition in Section 4.1 when we describe the datasets.

---

> > ### Author Response · Authors · 2023-03-13
> > **Thank you for your constructive criticism. (2/3)**
> >
> > - Reviewer fsHm: "(...) the authors say that the unsuccessful runs of the hyperparameter search pose a challenge for DEV, SND and ENT, but it is not clear why that is. (...)Please elaborate."
> >
> > With the wrong choice of hyper-parameters, we can end up with a degenerate model that predicts the same label for all examples with high confidence. This model will have good DEV, SND and ENT scores (being highly confident means it has low entropy) and therefore is considered a 'good' model according to these criterions when it's quite the contrary.
> >
> > --------------------------------
> >
> > - Reviewer fsHm: "Regarding hyperparameters, (...). This confused me for two reasons: first, isn’t the ‘end of training’ also a hyperparameter in and of itself (i.e the number of steps)? (...) Second, (...) This seems to contradict the statement of only considering the model at the end of training? (...)"
> >
> > Yes, the 'end of training' can be considered a hyper-parameter. We consider two phases in our study. Phase I consisted in finding the best hyper-parameters for each method. The length of training is fixed at 5000 steps for Office-Home and 10000 steps for VisDA. As we are looking for the best hyper-parameters based on a single task that hopefully generalize to the remaining tasks, by considering the end of training as a hyper-parameter we would run the risk of overfitting (anecdotally we found some tasks to converge faster than others). Therefore, in this Phase 1, we do not consider the 'end of training' as a hyper-parameter. In Phase 2, we take the chosen hyper-parameters and run the methods on all tasks. Here we use the model selection strategies to select the best model for each task along training, effectively considering the 'end of training' as a hyper-parameter.
> >
> > --------------------------------
> >
> > - Reviewer fsHm: "(...) The ‘with our neural network architecture’ part makes it sound like a different architecture was used for tuning rather than testing? (...) Please explain."
> >
> > See answer above. The 'with our own neural network architecture' is meant to distinguish the additional experiments with a different architecture that we later alude to. Results with different architectures were only reported in Table 4.
> >
> > --------------------------------
> >
> > - Reviewer fsHm: "Section 5 about reproducibility: (...) I wouldn’t really call this a reproducibility trial then, as the architecture is an important component which is changed. For the sake of verifying reproducibility / correctness of implementation, it would make more sense to use the same setup (...)"
> >
> > We agree with the reviewer and that is exactly what we did. Allow us to clarify. The reproducibility discussion is limited to Section 5.1. In it, we compare previously reported results to our results (see Table 4) using the same architectures as in the original works as well as the one used in our work. That is why we refer to it as "the reproducibility of previous results". In the rest of section 5, in order to study the model selection strategies, we use the same architecture for all methods in order to eliminate this factor of variations as the reviewer said.
> >
> > --------------------------------
> >
> > - Reviewer fsHm: "On a similar vein, it sounds like the authors have changed the architectures of some of the compared approaches, in order to have the same architecture used for all. (...)"
> >
> > Yes, we did change the architecture of SAFN and AR in comparison to their respective papers. The results in Table 11 show that the performance of both SAFN and AR dropped without the architecture changes. However, without further experiments, it is hard to speculate if the same architecture changes would also have a positive impact on the remaining methods. We find this to be an interesting question but leave it for future work.
> >
> > --------------------------------
> >
> > - Reviewer fsHm: "PADA combined with SND ‘performs better than the oracle on average, although within the standard deviation’: (...) I find the wording here confusing."
> >
> > We understand the reviewer's confusion and agree that in order to make a statement with statistical significance we would need to look into confidence intervals. We simply meant to say that the (PADA, SND) pair achieving **per run average**, while surprising at first, can be explained in part by the standard deviation. We updated the text to reflect that. It now reads:
> > "Perhaps a little surprising, when combined with SND it achieved a higher per run average than with ORACLE, although with the standard deviation. This is also}"
> >
> > --------------------------------
> >
> > - Reviewer fsHm: "Table 7: state what is the relationship between this and Table 6 (e.g. in the caption). Currently the captions of these two tables are similar and it takes some scrolling up and down to find the differences."
> >
> > Answer: Table 7 is the breakdown of the average task accuracy for VISDA displayed in Table 6 into the task specific average.

---

> > > ### Author Response · Authors · 2023-03-13
> > > **Thank you for your constructive criticism. (3/3)**
> > >
> > > - Reviewer fsHm: "Section 5.3, when discussing random seeds, please state what exactly these seeds control (...)"
> > >
> > > At the beginning of each run of each method, we set the seed of several different Python libraries following the recommendations in [1] in order to ensure that each run is reproducible.
> > >
> > > [1] https://pytorch.org/docs/stable/notes/randomness.html
> > >
> > > ---------------------------
> > >
> > > - Reviewer fsHm: "fix inconsistent notation, e.g. dev (in section 2) versus DEV (...)"
> > >
> > > Thank you for pointing it out. We have made the change so that Section 2 is consistent with the rest of the paper.
> > >
> > > ---------------------------
> > >
> > > - Reviewer fsHm: "Table 4 is hard to read as it contains a large number of figures. (...)?
> > >
> > > We considered the Reviewer's suggestion of using barplots, but found it wasn't the best medium to show the small differences between the reported results and our reimplementation. Instead, we have moved Table 4 into the Appendix/Supplemental Material (now table 11) and replaced it in the main paper with a new table with only the average task accuracy which is enough for our discussion in Section 5.1
> > >
> > >
> > > ---------------------------
> > >
> > > - Reviewer fsHm: "Regarding experimental setup, as explained in Section 4.1 for the datasets, can the particular selection of the first X categories to serve as the partial target domain bias the results? (...)?
> > >
> > > Choosing the first X categories to form the partial target domain goes back to Cao et al. (2018) who first described the PDA problem and proposed PADA. The community as a whole has adopted that choice and to best of our knowledge there is no work studying the impact of how choices of the categories to be included in the partial target domain impacts the results. It is possible that choosing the first X categories impacts the results, but we leave that for future work.

---

> > > > ### Comment · Reviewer_fsHm · 2023-03-20
> > > > **thank you for the responses**
> > > >
> > > > Thank you authors for the detailed responses and revisions.
> > > >
> > > > some quick follow-ups:
> > > >
> > > > - In the new paragraph added in Section 2, you say: "... be the labeled source (resp. target)" -- should it be "... be the labeled (resp. unlabeled) source (resp. target)", since the target is unlabeled.
> > > >
> > > > - I also find the sentence 'reduce the transfer of source-only samples' a bit confusing (what does the 'transfer of samples' mean?), though I do prefer it over negative transfer. Is what the authors mean here that it reduces the portion of target samples that are classified as belonging to source-only classes? If so, I think it would  be clearer to just say that.
> > > >
> > > > - 'although with the standard deviation' --> 'although within a standard deviation'?
> > > >
> > > > - I found your explanation of Phase I versus Phase II very helpful. It would be great to also add that to the revised paper (apologies if it's there and I missed it)

---

### Review · Reviewer_xx5V · 2023-03-03

**Summary Of Contributions:**

This work studies an important topic of model selection in Partial Domain Adaptation. In the current practice, there is no single established model selection technique and therefore each method adopts its own technique (often using target labels which violates the label-free assumption in Unsupervised Domain Adaptation).
The authors elucidate this challenge and develop a benchmarking framework - BenchmarkPDA with a focus on reproducibility and fair evaluation under a consistent setting. A rigorous analysis of several label-based (1-shot, 50-RND, 100-RND, ORACLE) and label-free (S-ACC, ENT, DEV, SND) model selection techniques is presented across various Partial DA methods, hyperparameters and seeds.
The key insight is that each DA method underperforms using label-free model selection techniques and labeled target data is necessary to select a model close to the oracle performance.

**Audience:**

Yes

**Broader Impact Concerns:**

This work investigates the reproducibility and stability of the existing partial domain adaptation methods and therefore does not impose any significant ethical concerns.

**Claims And Evidence:**

Yes

**Requested Changes:**

Overall, I found the paper interesting and addressing an important topic. This work can be further improved (please refer to the points in the Weaknesses section).

**Strengths And Weaknesses:**

Strengths:

* Readability: The manuscript is a good read - the writing is simple and clear and describes a spectrum of latest works in Partial DA and model selection strategies in Sec. 2 & 3.

* Significance: This study is of practical significance since model selection in Unsupervised DA is an important problem, and not commonly addressed by prior works. This work will motivate the study of more accurate model selection strategies that do not use expensive target annotations.

* Reproducibility: The BenchmarkPDA codebase is very useful for future research in PartialDA and encourages fair evaluation and reproducibility.

* Insights: Currently, the main goal is to reimplement, benchmark, and provide empirical results on the stability and reproducibility of the existing Partial DA methods using various model selection techniques. Though this has limited contributions in terms of algorithmic novelty, it is an important investigation which I believe has value to the progress of the field.

Weaknesses & scope for improvement:

* Motivation for PDA: The motivation to restrict the study to only Partial DA methods is unclear. As such the presented hypotheses could very well be applicable to other scenarios such as Open-set [P1], or Universal [P2] DA. Could the authors clarify why Partial DA was specifically chosen?

* Missing meta-insights: It would be great to understand if certain methods are more robust than others, and why. Here are some interesting questions which could lead to useful insights: "What contributes to the stability of  a method - is it due to less number of hyperparameters, or data-augmentation, or is it simply dataset and task dependent?", "Are there certain kinds of Partial DA approaches that are more reliable?", "Can we calibrate the trade-off between the amount of labeled examples required, and the reliability of model selection?", "What should a researcher keep in mind while developing a new model selection technique?". Building a narrative that answers meta-questions like these could strengthen the work.

* Minor comments:

  1) The second bullet point in Page 2 (“Only 1 pair…”) is not revisited later in the paper (did I miss it?). Also, point (iii) in Conclusion (“An ablation study…”) is a generic recommendation which is not talked about in the main text. The manuscript can be improved by providing a more comprehensive discussion on these points.

  2) Please fix typos:

    - Table 7 best and worst models are incorrectly highlighted: For S2R method SAFN, SND produces the worst model (not ENT), and for the method AR, ENT selects the best model.

    - Supplementary Table 6: In Office-Home dataset, for JUMBOT, the 50-RND selects worse models (10% worse) than 1-SHOT and 100-RND which looks like a typo. Please double check the same.

  3) "7 different state-of-the-art Partial DA algorithms": In my opinion, the source-only model should not be considered as a partial DA algorithm. Thus there are six PDA algorithms that are studied in this work (PADA, SAFN, BA3US, AR, JUMBOT, MPOT). I would suggest the authors to make appropriate changes to reflect the same.

References:

[P1] Cao et al., "Separate to Adapt: Open Set Domain Adaptation via Progressive Separation", CVPR 2019

[P2] Yu et al., "Universal Domain Adaptation", CVPR 2019

---

> ### Author Response · Authors · 2023-03-13
> **Thank you for your positive comments (1/2)**
>
> We would like to thank the reviewer for the positive comments especially about the rigour of our empirical study. We also thank the reviewer the detailed review and the time taken on our manuscript. Below we answer each of the raised concerns. We have updated our manuscript and all change can be seen in blue.
>
>
> - Reviewer xx5V: "Motivation for PDA."
>
> The reviewer makes an interesting point. It is true that several other domain adaptation variants could have been considered. However, we would like to point out that the standard domain adaptation setting has been studied in concurrent work [1]. Furthermore, label shift between domains is typical in real-world data. The partial domain adaptation setting corresponds to an extreme label shift scenario where some source classes are not in the target domain, and it needs dedicated solutions as typical domain adaptation methods do not perform well on the PDA setting. More generally, other DA variants also require dedicated solutions to handle the considered DA scenario. We believe that it would have been hard to test several methods with several model selection strategies in the same paper for all domain adaptation variants. That is why we prefered tackling only one domain adaptation setting. We have added a paragraph in the introduction explaining our choice.
>
> On a final note, addressing the other variants would have required significantly more experiments which we are unable to tackle with the computational resources available to us (8 NVIDIA GeForce RTX 2080 Ti). Thus, we decided to focus on the PDA variant given our expertise in this domain and given that previous work that proposes model selection strategies ([2, 3]) focus its attention on standard domain adaptation claiming their strategies to work on PDA but doing so based on limited experiments.
>
>
> [1] Musgrave et al, Three New Validators and a Large-Scale Benchmark Ranking for Unsupervised Domain Adaptation
> [2] You et al, Towards Accurate Model Selection in Deep Unsupervised Domain Adaptation
> [3] Saito et al, Tune it the Right Way: Unsupervised Validation of Domain Adaptation via Soft Neighborhood Density
>
> -----------------------------
>
> - Reviewer xx5V: "Missing meta-insights"
>
> We agree with the reviewer that proving such meta-insights would strengthen the paper. We answer to each of the concern below
>
> "What contributes to the stability of a method - is it due to less number of hyperparameters, or data-augmentation, or is it simply dataset and task dependent?"
> In order to address the stability of the method, we focus our attention on the results in Table 1 which allows us to compare the best and worst performance without target labels for each method, we see that PADA and BA3US are not robust to the choice of model selection strategy for either one of the datasets, while SAFN is robust to the choice of model selection strategy for Office-Home. These are the methods with the fewest hyper-parameters ruling out the fact that fewer hyper-parameters lead to more robust methods and suggesting it is method and dataset specific. We added this discussion in Section 5.2.
>
> "Are there certain kinds of Partial DA approaches that are more reliable?"
> Regarding reliability, we find the optimal transport based approaches (JUMBOT and MPOT) to be the most sensitive to model selection strategies as they exhibit the largest performance gaps between the best and worse model selection strategies without target labels. It shows, that while they are able to achieve SOTA results with the Oracle model selection strategy, that performance is highly tied to the hyper-parameter choice. We added this discussion in Section 5.2.
>
> "Can we calibrate the trade-off between the amount of labeled examples required, and the reliability of model selection?"
> From our experiments, we found that 100 labelled samples are needed to achieve results close to Oracle and there is a noticeable, albeit small, drop in performance in using 50 labelled samples. This was mentioned in Section 5.2 of our manuscript.
>
> "What should a researcher keep in mind while developing a new model selection technique?"
> We believe model selection techniques should be fully tested in choosing all hyper-parameters of the method instead of simply deciding when to stop training.
>
> -----------------------------
>
> - Reviewer xx5V: "(minor comments) The second bullet point in Page 2 (“Only 1 pair…”) is not revisited later in the paper (did I miss it?).
>
> We indeed mentioned it in the paragraph Model Selection Strategies (w/o target labels) in Section 5. We have added details to this part in order to make this point clearer.
>
> -----------------------------
>
> - Reviewer xx5V: "Also, point (iii) in Conclusion (“An ablation study…”) is a generic recommendation which is not talked about in the main text. The manuscript can be improved by providing a more comprehensive discussion on these points."
>
> Thank you for your suggestion. We have added a discussion on ablation studies in Section 5.1.

---

> > ### Author Response · Authors · 2023-03-13
> > **Thank you for your positive comments (2/2)**
> >
> > - Reviewer xx5V: : "Table 7 best and worst models are incorrectly highlighted(...)
> > Supplementary Table 6: In Office-Home dataset, for JUMBOT, the 50-RND selects worse models (10% worse) than 1-SHOT and 100-RND which looks like a typo. (...).
> > (...) In my opinion, the source-only model should not be considered as a partial DA algorithm. (...)"
> >
> > Thank you for your point about the different typos found in our manuscript. We have made a careful reading and have considered your remarks in our updated manuscript. Interestingly, the accuracy of JUMBOT with the 50-RND validation set was not a typo. When using a small set of labelled samples, the target accuracy estimator is a poor estimator and may lead to overestimates. This is precisely what happened with the JUMBOT with the 50-RND where the use of 50-RND led to the very suboptimal choice of hyper-parameters explaining its poor performance.

---

> ### Comment · Reviewer_xx5V · 2023-03-27
> **Thanks for the responses**
>
> Thanks to the authors for the responses; my broad concerns are addressed.
>
> Quick comment -- the abstract reads "7 state-of-the-art PDA algorithms", please update it to "6" in the final version.
>
> Overall, I find the work quite interesting and I appreciate the revisions that have improved the insights and the clarity of the manuscript.

---

> > ### Author Response · Authors · 2023-03-27
> > **Thank you for your answer.**
> >
> > We have corrected the typo about the number of PDA algorithms. We will make a careful final reading to check the remaining typos.
> >
> > We would like to thank you for the discussion and we have appreciated your constructive feedback and criticisms.
> >
> > Best regards,
> >
> > The authors.

---

### Review · Reviewer_6NSE · 2023-03-04

**Summary Of Contributions:**

This paper provides a comprehensive investigation to existing partial domain adaptation methods with various model selection strategies. It’s interesting to know that existing PDA methods failed at an UDA setting while with 100 labelled test samples can ensure an effective PDA.

**Audience:**

Yes

**Claims And Evidence:**

Yes

**Requested Changes:**

See the weakness.

**Strengths And Weaknesses:**

There are some minor issues that should be addressed for better readability and soundness.

It’s misleading to claim that PDA methods decrease up to 30% percentages according to Table 1. Please revise.

Also in Table 1, besides of reporting the worst and best results on different random seed, please also report the average results with variances.

In the first paragraph, domain shift is clearly not only limited to different backgrounds or colours. Therefore, ‘such shift is referred to as domain shift in the literature’ is not accurate.

at the end of first paragraph, ‘outlier source only label’ may not be accurate. I understand because source only samples may be regarded as outliers w.r.t target distribution. Please rewrite it to avoid confusion as normally outlier is used w.r.t training set.

In table 2 row Architecture, the caption should clarify what does the symbol between networks mean. A table should explain itself.

---

> ### Author Response · Authors · 2023-03-13
> **Thank you for your constructive review.**
>
> We would like to thank the reviewer for taking the time to review our manuscript and the encouraging comments. Below we address each one of your comments. We highlight the changes in our updated manuscript in blue.
>
> - Reviewer 6NSE: "It’s misleading to claim that PDA methods decrease up to 30% percentages according to Table 1. Please revise."
>
> We understand the confusion in our claim and we have revised it as follows: "For a given method, the difference of accuracy attained by a model selected without target labels and a model selected with target labels can reach 30 percentage points". See our updated manuscript.
>
> -----------------
> - Reviewer 6NSE: "Also in Table 1, besides of reporting the worst and best results on different random seed, please also report the average results with variances."
>
> Thank you for your suggestion. We have updated the table to include as well the average case and the variances.
>
> -----------------
>
> - Reviewer 6NSE: "In the first paragraph, domain shift is clearly not only limited to different backgrounds or colours. Therefore, ‘such shift is referred to as domain shift in the literature’ is not accurate."
>
> We agree with the reviewer and we have rewritten this sentence in intro.
>
> -----------------
>
> - Reviewer 6NSE: "at the end of first paragraph, ‘outlier source only label’ may not be accurate. I understand because source only samples may be regarded as outliers w.r.t target distribution. Please rewrite it to avoid confusion as normally outlier is used w.r.t training set."
>
> We have reworded this discussion in the manuscript without using of the word outlier as requested.
>
> -----------------
>
> - Reviewer 6NSE: "In table 2 row Architecture, the caption should clarify what does the symbol between networks mean. A table should explain itself."
>
> Answer : Thank you for pointing it out. We have updated the caption of Table 2 accordingly.

---

### Decision · Action_Editors · 2023-05-08

**Recommendation:** Accept as is

**Comment:**

This is more a survey/benchmark paper, so the comprehensiveness and elaboration are the crucial factors impacting the paper quality. In a nutshell, this paper has great work in achieving these. The outcome of this paper is a solid BenchmarkPDA for reproducibility, and insights about the robustness and stability of well-established partial domain adaptation methods, with particular insights into the validation and parameter tuning strategies. Three reviewers assessed this paper and the revisions, with active interactions with the authors through OpenReview. Authors managed to address all comments reasonably well, which leads to unanimous acceptance recommendations from the reviewers. The Associate Editor agrees that this is a good paper that is meaningful for the community, and thus accepts the paper as is.

**Audience:**

Yes. Researchers working on domain adaptation, domain generalization, and open-set recognition, to name a few, would have interests in this paper.

**Claims And Evidence:**

Yes. This paper provides a solid benchmark with helpful insights towards the evaluation of partial domain adaptation methods.